# Telomeres control human telomerase (*TERT*) expression through non-telomeric TRF2

**Antara Sengupta[1,2,3†], Soujanya Vinayagamurthy[1,2,3], Dristhi Soni[1,2,3‡], Rajlekha Deb[1,3], Ananda Kishore Mukherjee[1,2,3§], Subhajit Dutta[1,2,3], Jushta Jaiswal[1,3#], Mukta Yadav[1,2,3], Shalu Sharma[1,2,3¶], Sulochana Bagri[1,2,3], Shuvra Shekhar Roy[1,2,3], Priya Poonia[1,3], Ankita Singh[1,3], Divya Khanna[1,3], Amit Kumar Kumar Bhatt[1,3], Akshay Sharma[4], Suman Saurav[4], Rajender K Motiani[4], Shantanu Chowdhury[1,2,3*]**

[1]Integrative and Functional Biology Unit, CSIR-Institute of Genomics and Integrative Biology, Delhi, India; [2]Academy of Scientific and Innovative Research (AcSIR), Ghaziabad, India; [3]CSIR-Institute of Genomics and Integrative Biology, New Delhi, India; [4]Laboratory of Calciomics and Systemic Pathophysiology (LCSP), Regional Centre for Biotechnology (RCB), Faridabad, India

**\*For correspondence:**
shantanuc@igib.in

**Present address:** †Royal College of Surgeons in Ireland, Dublin, Ireland; ‡McGill University, Montreal, Canada; §Kennedy Institute of Rheumatology, University of Oxford, Oxford, United Kingdom; #School of Life Sciences, University of Warwick, Coventry, United Kingdom; ¶National Cancer Institute, Radiation Oncology Branch, Bethesda, United States

**Competing interest:** The authors declare that no competing interests exist.

## eLife Assessment

The authors of this **important** study investigate how telomere length regulates hTERT expression via non-telomeric binding of the telomere-associated protein TRF2. They conclusively show that TRF2 binding to long telomeres results in a reduction in its binding to the hTERT promoter, while short telomeres restore TRF2 binding in the hTERT promoter, recruiting repressor complexes like PRC2, and suppressing hTERT expression. There is **convincing** support for the claims and the findings should be of broad interest for cell biologists and those working in fields where telomeres alter function, such as cancer and aging.

**Abstract** The function of the human telomerase reverse transcriptase (referred hereafter as *TERT*) in the synthesis and maintenance of chromosome ends, or telomeres, is widely understood. Whether and how telomeres, on the other hand, influence *TERT* regulation is relatively less studied. We found *TERT* was transcriptionally altered depending on telomere length (TL). This resulted from TL-dependent binding of TRF2 between telomeres and the *TERT* promoter. *TERT* promoter-bound TRF2 was non-telomeric and did not involve the looping of telomeres to the *TERT* promoter. Cell lines from different tissue types fibrosarcoma (HT1080), colon cancer (HCT116), and breast cancer (MDA-MB-231), engineered for either telomere elongation/shortening, gave an increase/decrease in *TERT*, respectively. Mechanistically, we show *TERT* promoter-bound non-telomeric TRF2 recruits the canonical PRC2-complex, inducing repressor histone H3K27-trimethylation in a TL-dependent fashion. This was further supported by TL-dependent promoter activity from an exogenously inserted *TERT* reporter. Increase in TL over days followed by a gradual decline, resulted in activation followed by repression of *TERT* in a concerted manner, further implicating TL as a key factor for *TERT* regulation. Notably, on reprogramming primary fibroblasts to induced pluripotent stem cells (iPSCs), TRF2 loss from the *TERT* promoter was evident along with telomere elongation and *TERT* upregulation. Conversely, on telomere shortening in iPSCs, *TERT* promoter-bound TRF2 was restored with a marked reduction in *TERT*, further supporting the causal role of TL in *TERT* transcription. Mechanisms of tight control of *TERT* by

TL shown here are likely to have major implications in telomere-related physiologies, particularly, cancer, ageing, and pluripotency.

## Introduction

Telomeres are nucleoprotein structures comprising guanine-rich DNA sequences at the end of linear chromosomes (*Rhodes et al., 2002*; *O'Sullivan and Karlseder, 2010*; *Shay and Wright, 2019*). The catalytic subunit of the ribonucleoprotein telomerase, human telomerase reverse transcriptase (*TERT*) is necessary for replicating telomeric repeats to maintain telomeres (*Greider and Blackburn, 1985*; *Blackburn, 2001*; *Autexier and Lue, 2006*; *Smith et al., 2020*). Most cancer cells, in contrast to normal adult somatic cells, maintain telomeres through reactivation of *TERT* (~90% of cancers); alternative lengthening of telomeres (ALT) is used in other cases (*Shay and Bacchetti, 1997*; *Pandita et al., 2015*; *Heaphy et al., 2011*; *Akincilar et al., 2016b*; *Sharma and Chowdhury, 2022*). *TERT*, therefore, is tightly regulated at the epigenetic, transcriptional, and translational levels (*Roake and Artandi, 2020*; *Cong et al., 2002*).

The role of *TERT* in the synthesis and maintenance of telomeres has been extensively studied (*Shay and Wright, 2019*; *Blackburn, 2001*; *Smith et al., 2020*; *Roake and Artandi, 2020*; *Sarthy and Baumann, 2010*; *D'Souza et al., 2013*). Relatively recent work suggests how telomeres, on the other hand, might affect *TERT* regulation. Physical looping of telomeres, reported as the Telomeric Position Effect-over-long-distance (TPE-OLD) was shown to extend up to 10 Mb of telomeres (subtelomeric). The resulting telomeric heterochromatinisation gave TL-dependent repression of subtelomeric genes. The *TERT* loci (~1.2 Mb away from telomeres) was shown to be repressed through (TPE-OLD) (*Robin et al., 2014*; *Kim et al., 2016*; *Kim and Shay, 2018*) loss of looping in short telomeres induced *TERT*. Notably, another conceptually distinct mechanism of TL-dependent gene regulation was reported, which influenced genes spread throughout the genome: expression of genes distal from telomeres (for instance, 60 Mb from the nearest telomere) was altered in a TL-dependent way, but without physical telomere looping interactions. First, non-telomeric binding of TRF2 was found to be extensive (>20,000 sites), spread across the genome, including promoters, and affected the epigenetic regulation of genes (*Mukherjee et al., 2018b*; *Mukherjee et al., 2019*; *Purohit et al., 2018*). Second, the shortening or elongation of telomeres led to the release or sequestration of telomeric TRF2, respectively, thereby increasing or decreasing the availability of TRF2 at non-telomeric promoters and affecting gene expression (*Mukherjee et al., 2018b*; *Mukherjee et al., 2021*). The telomeric sequestration-dependent partitioning of telomeric versus non-telomeric TRF2 was proposed as the Telomere-Sequestration-Partitioning (TSP) model, linking global gene regulation to telomeres (*Figure 1*; *Mukherjee et al., 2018b*; *Mukherjee et al., 2018a*; *Okamoto and Seimiya, 2019*; *Vinayagamurthy et al., 2020*). Further, we found promoter-bound TRF2, not associated with telomeres (or non-telomeric), repressed *TERT* (*Sharma et al., 2021*). Based on these, here we tested the hypothesis that non-telomeric TRF2 binding at the *TERT* promoter was TL-dependent. If so, this would molecularly link telomeres to *TERT* regulation.

Findings that show TRF2 binding at the *TERT* promoter increased with telomere shortening and downregulated *TERT*. Conversely, reduced promoter TRF2 on telomere elongation upregulated *TERT*; consistent with TSP described above. This was clear in multiple cell types engineered for either telomere elongation or shortening. An artificially inserted *TERT* promoter (>40 Mb from telomere) showed TRF2-dependent and telomere-sensitive regulation, confirming the mechanism to be independent of physical telomere looping. For a temporal model, we elongated telomeres using doxycycline (dox) inducible cells and stopped elongation after a desired time. TRF2 binding at the *TERT* promoter decreased as telomeres elongated over several days, and increased again with telomere shortening when dox treatment was stopped. Together, these show a heretofore unknown mechanism of telomere-dependent regulation of *TERT*, the sole enzyme necessary for synthesizing telomeres, revealing a cellular feedback machinery connecting telomeres and *TERT* expression.

**eLife digest** Human cells typically have 23 pairs of structures known as chromosomes, each containing a unique set of genes that provides the instructions needed to make proteins and other essential molecules found in the body.

At the end of each chromosome lies a region of repetitive DNA sequences, known as the telomere, which protects the chromosome strands from damage or tangling. Every time a cell divides, some of the telomere repeats are cut off, causing the telomeres to shorten. Specific enzymes known as telomerases can add these repeats back on so that the telomeres do not become too short.

As we age, telomeres naturally shorten in length, but certain lifestyle factors can accelerate this process, leading to programmed cell death and contributing to various diseases, including cancers.

In 2018, researchers showed that TRF2, a key telomere protein, helps relay information about telomere length to various regulatory genes. However, it remained unclear whether TRF2 could directly communicate this information to the telomerase itself. Sengupta et al. – including several of the researchers from the 2018 study – set out to answer this question by studying multiple human cancer cell lines with different telomere lengths.

They discovered that TRF2 acts like a messenger that interacts with the telomerase gene *TERT*, in a length-dependent manner. When telomeres were long, most TRF2 remained bound at the telomeres and did not interact with *TERT*, allowing continued telomerase activity and telomere elongation. By contrast, when telomeres were short, TRF2 was released from the telomeres and bound directly to the *TERT* promoter. There, together with a DNA structure called the G-quadruplex, TRF2 suppressed *TERT* expression and thereby limited further telomere extension. In other words, shortened telomeres repressed *TERT* expression, while elongated ones increased it.

In this positive feedback–like reinforcement system, telomere length is signaled by the amount of free TRF2 protein. When TERT expression is high and telomeres are long, less TRF2 is available to bind the *TERT* promoter, reducing repression and thereby maintaining its expression. Conversely, when TERT expression is low and telomeres are short, more TRF2 is available, reinforcing repression and sustaining the low-expression state.

Regulating telomere length is essential for healthy cell function, and its dysregulation is a hallmark of cancer. Sengupta et al. demonstrated that TRF2 plays a critical role in this process by feeding back information about telomere length directly to the TERT gene. By further dissecting this feedback system and refining experimental tools, researchers may gain new insights into telomere regulation in cancer and related diseases.

## Results

### Telomere elongation upregulates *TERT* through non-telomeric TRF2

To test if TL of cells affected TRF2 occupancy at the *TERT* promoter, we first elongated telomeres in cancer cells under isogenic backgrounds (termed as S̲hort T̲elomere, ST, or L̲ong T̲elomere, LT cells hereon). We used HT1080 fibrosarcoma cells (HT1080-ST) or the TL-elongated version HT1080-LT (with constitutive expression of *TERT* and the RNA component *TERC*; previously characterized by others and us, see Methods) (*Mukherjee et al., 2018b*; *Cristofari and Lingner, 2006*). TL difference between HT1080-ST/LT was confirmed by flow cytometry (FACS, *Figure 2A* and qPCR-based method, *Figure 2—figure supplement 1A*, and telomerase activity in *Figure 2—figure supplement 1B*). Chromatin-immunoprecipitation (ChIP) of TRF2 gave significantly reduced occupancy of TRF2 on the *TERT* promoter in HT1080-LT compared to HT1080-ST cells (*Figure 2B*, ChIP-qPCR spanning 0–300 bp upstream of transcription start site, TSS).

TRF2-dependent H3K27 trimethylation (*Mukherjee et al., 2018b*), including at the *TERT* promoter (*Sharma et al., 2021*), was reported earlier by us: TRF2 silencing reduced the H3K27me3 mark at the *TERT* promoter along with an increase in *TERT* expression in the different cell lines tested (*Sharma et al., 2021*). Here, we asked if TRF2-dependent *TERT* regulation was TL-sensitive. With TL elongation, the H3K27me3 mark was significantly reduced in HT1080-LT relative to the ST cells (*Figure 2C*). As expected from reduced promoter H3K27 trimethylation, *TERT* transcription was enhanced

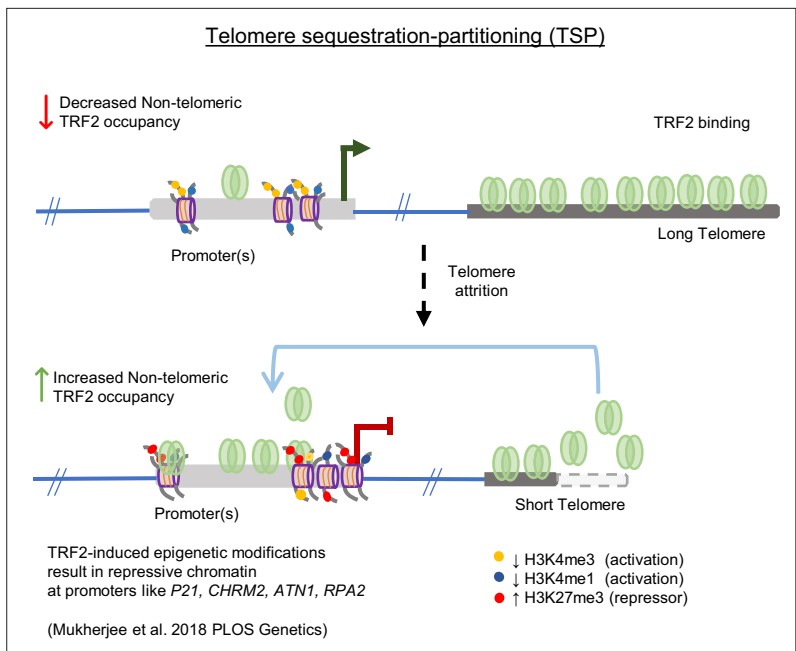

**Figure 1.** Graphical representation of the Telomere Sequestration Partitioning (TSP) model. The binding of TRF2 outside of telomeric regions (non-telomeric) is dependent on telomere length. When telomeres shorten, the occupancy of TRF2 at telomeres diminishes, resulting in increased TRF2 binding at non-telomeric sites. This shift in TRF2 distribution triggers epigenetic modifications at promoters showing telomere-dependent gene regulation. Schematic reused from Supplementary Information in Mukherjee et al, eLife, 2025.

by >threefolds in HT1080-LT relative to ST cells; 3'UTR-specific primers were used to distinguish from exogenously expressed *TERT* (*Figure 2D*).

To rule out any effect of exogenous *TERT* (HT1080-LT cells above) we sought to elongate telomeres in a *TERT*-independent way. Using a reported method, telomeres were elongated in MDA-MB-231 breast cancer cells with guanine-rich telomeric repeat (GTR) oligonucleotides characterized by others and us, see Methods; termed MDA-MB-231-ST/LT; TL difference quantified using FACS, *Figure 2A*, and qPCR-based method, *Figure 2—figure supplement 1*, and telomerase activity in ; *Mukherjee et al., 2018b*; *Wright et al., 1996*. MDA-MB-231-LT cells had lower TRF2 binding at the *TERT* promoter compared to MDA-MB-231-ST cells (*Figure 2B*). Accordingly, H3K27 trimethylation at the *TERT* promoter was reduced and *TERT* upregulated by ~threefolds in MDA-MB-231-LT compared to MDA-MB-231-ST cells (*Figure 2C–D*). Together, TL-sensitive regulation of *TERT* was consistent in both HT1080 and MDA-MB-231 cells on telomere elongation.

## Telomere shortening increased promoter TRF2 suppressing, *TERT* expression

As an alternative model, instead of elongation, we shortened TL in HCT116 colorectal carcinoma cells with a higher average TL (~5.6 kb) compared to HT1080 (~4.5 kb) or MDA-MB-231 (~3 kb) cells (*Mukherjee et al., 2018b*; *Cristofari and Lingner, 2006*; *Myung et al., 2004*). Two different methods were used for TL shortening:

(a) *TERT*-independent, telomere-specific sgRNA guided CRISPR-Cas9 to trim telomeres (see Methods, scheme in *Figure 2—figure supplement 1C*) or (b) depletion of the *TERC* RNA (see Methods, TLs in HCT116-ST/LT quantified using FACS; *Figure 2A*, and qPCR based method, *Figure 2—figure supplement 1*, and telomerase activity in *Figure 2—figure supplement 1*). The ST cells in both TL-shortened models had higher TRF2 binding on the *TERT* promoter compared to the HCT116-LT cells (*Figure 2B*). Also, we found a higher repressor H3K27me3 mark at the *TERT* promoter, as expected from increased TRF2, in ST compared to its LT counterpart in both the HCT116 models (*Figure 2C*). As expected, an increase in the H3K27me3 repressor mark at the *TERT* promoter resulted in > twofold reduced *TERT* expression in ST relative to the HCT116-LT cells (*Figure 2D*).

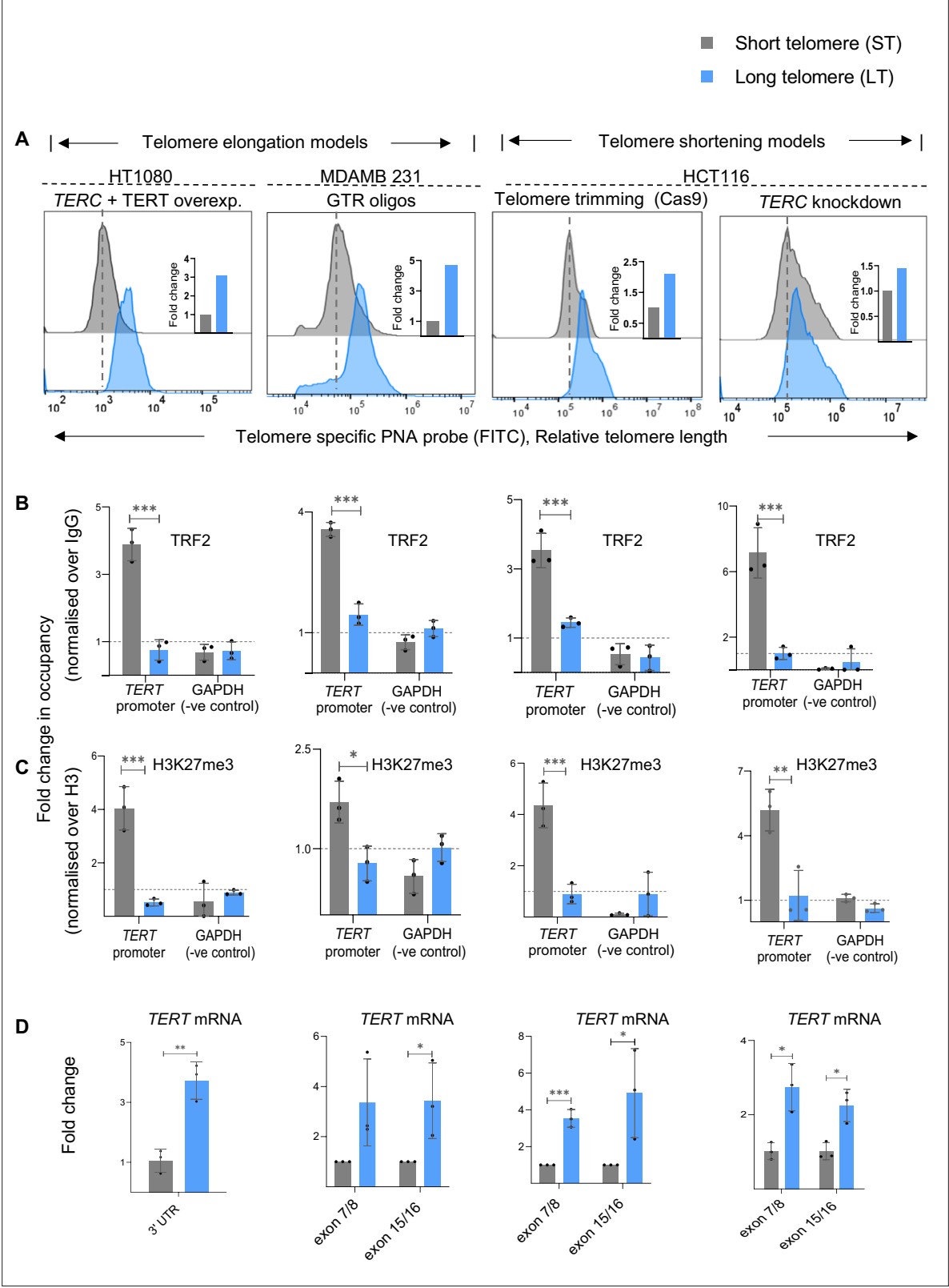

**Figure 2.** Telomere-dependent non-telomeric TRF2 binding at the telomerase reverse transcriptase (*TERT*) promoter controls *TERT* expression. (**A**) Telomere length in isogenic cancer cell lines with short telomeres (ST, in grey) or long telomeres (LT, in blue) namely, HT1080-ST/LT, MDA-MB-231-ST/LT, and HCT116- ST/LT (Telomere trimming-Cas9 and *TERC* knockdown) as determined by Flow cytometry (FACS). MDA-MB-231-ST/LT and HCT116-ST/LT Telomere trimming-Cas9 models were generated by telomerase-independent mode of telomere length (TL) alteration (See Methods).

*Figure 2 continued on next page*

*Figure 2 continued*

Relative fold change is shown in insets with FACS plots. (**B–C**) ChIP followed by qRT-PCR at the 0–300 bp *TERT* promoter (upstream of TSS) for TRF2 (**B**) and H3K27me3 (**C**) in respective ST/LT cells as mentioned in (**A**); occupancy normalised to respective IgG or total Histone H3 (for H3K27me3). qPCR on the *GAPDH* promoter was used as the negative control in all cases. (**D**) *TERT* mRNA expression by qRT-PCR in ST/LT cell line pairs as mentioned in (**A**) normalised to *GAPDH* or 18 S mRNA levels. Primers specific to 3'UTR for endogenous *TERT* were used for the HT1080-ST/LT system where telomerase was overexpressed for telomere elongation; primers for functional (reverse transcriptase domain) (exon7/8) and full-length (exon 15/16) transcript were used for all other systems. MDA-MB-231-ST/LT and HCT116-ST/LT Telomere trimming-Cas9 models have been analysed as paired samples in each biological replicates. Error bars represent ± SDs from the mean of three independent biological replicates of each experiment. P-values are calculated by an unpaired t-test for all data except mRNA for MDA-MB-231-ST/LT and HCT116-ST/LT Telomere trimming-Cas9 models where paired t-test was performed. (*p<0.05, **p<0.01, ***p<0.005, ****p<0.0001).

The online version of this article includes the following source data and figure supplement(s) for figure 2:

**Source data 1.** Source data for all plots in *Figure 2*, related to HT1080 ST/LT cell line model.

**Source data 2.** Source data for all plots in *Figure 2*, related to MDA MB 231 ST/LT cell line model.

**Source data 3.** Source data for all plots in *Figure 2*, related to HCT116 ST/LT Telomere trimming-Cas9 model.

**Source data 4.** Source data for all plots in *Figure 2*, related to HCT116 ST/LT *TERC* knockdown model.

**Figure supplement 1—source data 1.** Source data for all plots in *Figure 2—figure supplement 1*.

**Figure supplement 1.** Telomere-dependent non-telomeric TRF2 binding at the telomerase reverse transcriptase (*TERT)* promoter controls *TERT* expression (continued).

Previously, enhanced H3K27 trimethylation from TRF2-dependent binding of the RE1- silencing factor (REST) and EZH2 (the catalytic component of the PRC2 repressor complex) was reported at the *TERT* promoter (*Sharma et al., 2021*). Here, we asked if the TL-sensitive H3K27 trimethylation resulted from altered REST/EZH2 binding at the *TERT* promoter. ChIP of REST showed enhanced occupancy at the *TERT* promoter in ST relative to HT1080-LT cells (*Figure 3A*). EZH2 binding was also higher at the *TERT* promoter in ST compared to HT1080-LT cells (*Figure 3B*). Similarly, in MDA-MB-231 cells, REST and EZH2 occupancy at the *TERT* promoter was lower in LT compared to ST cells (*Figure 3A–B*). Furthermore, an increase in REST and EZH2 binding at the *TERT* promoter in ST relative to LT cells was clear in the HCT116-ST/LT cells (*Figure 3A–B*).

## Artificially inserted *TERT* promoter is telomere sensitive

To further test TL-sensitive *TERT* promoter activity, we used a reporter cassette comprising 1300 bp of the *TERT* promoter upstream of the *Gaussia* luciferase (*Gaussia* Luc) gene. The reporter was inserted at the CCR5 safe-harbour locus, 46 Mb away from the nearest telomere, using CRISPR in HEK293T cells (*Figure 4A*). TL in HEK293T cells was shortened using telomere-specific sgRNA guided CRISPR-Cas9 to trim telomeres *Figure 4B*, see Methods, scheme in *Figure 2—figure supplement 1*. Higher TRF2 binding at the inserted *TERT* promoter in ST compared to the HEK293-LT cells was clear (ChIP-qRT primers specific to the CCR5-*TERT* insert were used; *Figure 4A and C*). Consistent with this, H3K27me3 deposition was relatively high in ST cells (*Figure 4D*), and the *TERT* promoter-driven *Gaussia* Luc reporter activity was reduced in the ST relative to the LT cells (*Figure 4E*). Together, these show TL-dependent *TERT* regulation through non-telomeric TRF2 as a likely intrinsic mechanism.

## Temporally elongated or shortened telomeres up or downregulate *TERT*

We next studied the effect of time-dependent (temporal) TL elongation/shortening, i.e., TL increase followed by a decrease in a continuous way. For this, we made stable HT1080 cells with doxycycline-inducible *TERT*. Following induction of *TERT,* and telomerase activity (Day 0), we checked TL through days 6/8/10/16 and 24 (*Figure 5—figure supplement 1A*). Gradual increase in TL to > fivefold by Day 10 (relative to Day 0) was evident; beyond Day 10, despite dox induction (and relatively high telomerase activity) further TL elongation was not seen (*Figure 5—figure supplement 1A–B*; ++/-- denotes presence/absence of dox, respectively).

For detailed analysis, we focused on days 0,10, and 24 as representative time points; with discontinuation of dox after Day 10 (Dox-HT1080, *Figure 5A*). Following the withdrawal of dox after Day 10, TL returned to roughly within 1.6-fold of the initial (Day 0) state by Day 24 (*Figure 5B*, *Figure 5—figure supplement 1A*).

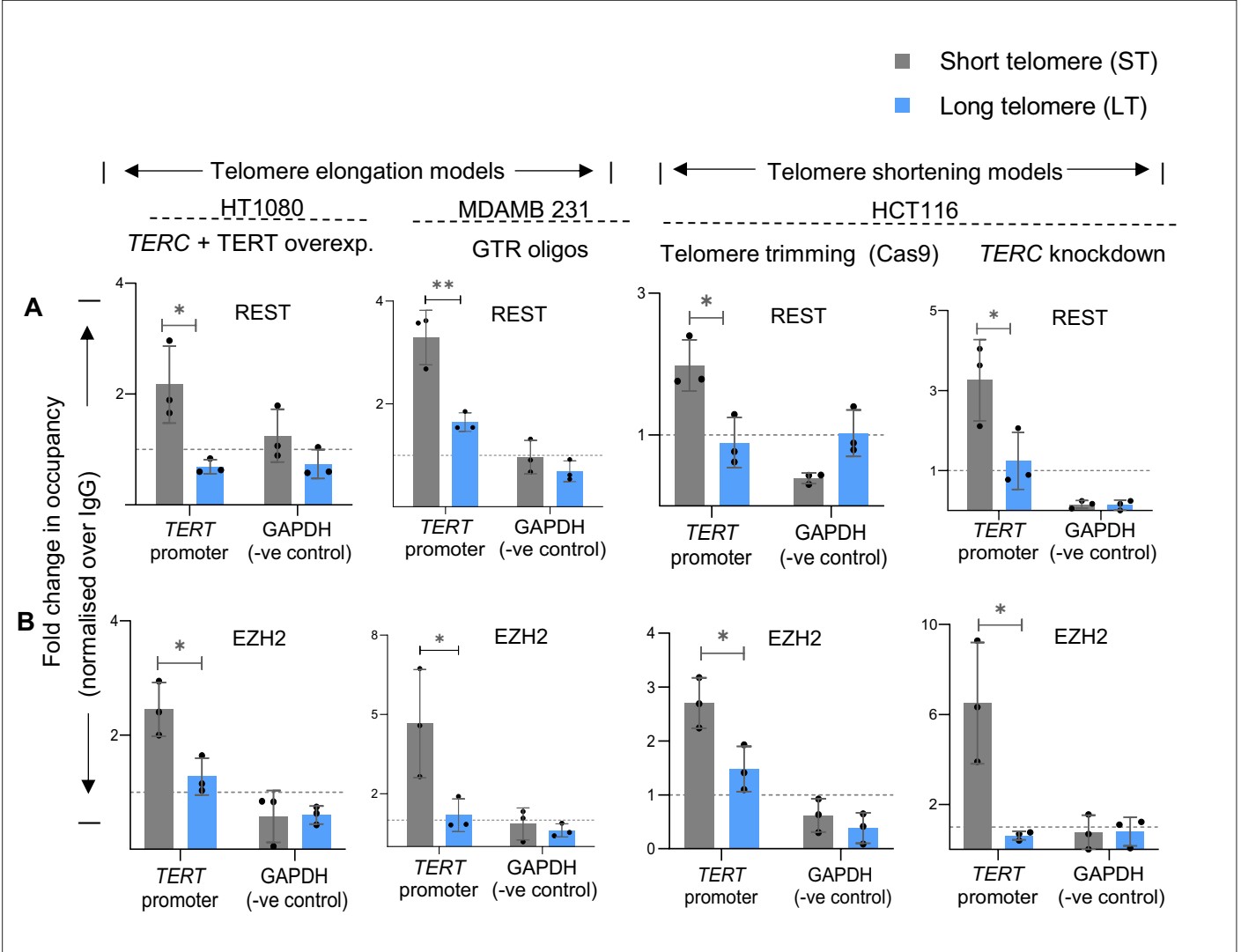

**Figure 3.** Telomere length dictates chromatin accessibility at the telomerase reverse transcriptase (*TERT)* promoter. (**A**, **B**) ChIP followed by qRT-PCR at the *TERT* promoter for REST (**A**) and EZH2 (**B**) in HT1080-ST/LT, MDA-MB-231-ST/LT, and HCT116- ST/LT (Telomere trimming-Cas9 and *hTERC* knockdown) cells; occupancy normalised to respective IgG. qPCR on the *GAPDH* promoter was used as the negative control in all cases. Error bars represent ± SDs from the mean of three independent biological replicates of each experiment. P-values are calculated by unpaired t-test (*p<0.05, **p<0.01, ***p<0.005, ****p<0.0001).

The online version of this article includes the following source data for figure 3:

**Source data 1.** Source data for all plots in *Figure 3*.

TRF2 binding at the *TERT* promoter decreased at Day 10 with TL elongation. Conversely, as TL shortened beyond Day 10, TRF2 occupancy on the *TERT* promoter was regained on Day 24; *Figure 5C*. Notably, this showed TRF2 binding at the *TERT* promoter to be temporally controlled as a function of TL. Second, this further showed that *TERT* promoter-bound TRF2 was non-telomeric because telomere-associated TRF2 would increase with TL elongation (and looping), instead of decreasing as seen here.

We then examined the occupancy of REST and EZH2 at the *TERT* promoter at days 0, 10, and 24: a decrease in occupancy of both REST and EZH2 on Day 10, relative to Day 0, followed by an increase on Day 24 was clear (*Figure 5D and E*).

Accordingly, H3K27me3 deposition at the *TERT* promoter was lower on Day 10 compared to Day 0, followed by an increase on Day 24 (*Figure 5F*). Taken together, clearly, REST and EZH2 binding,

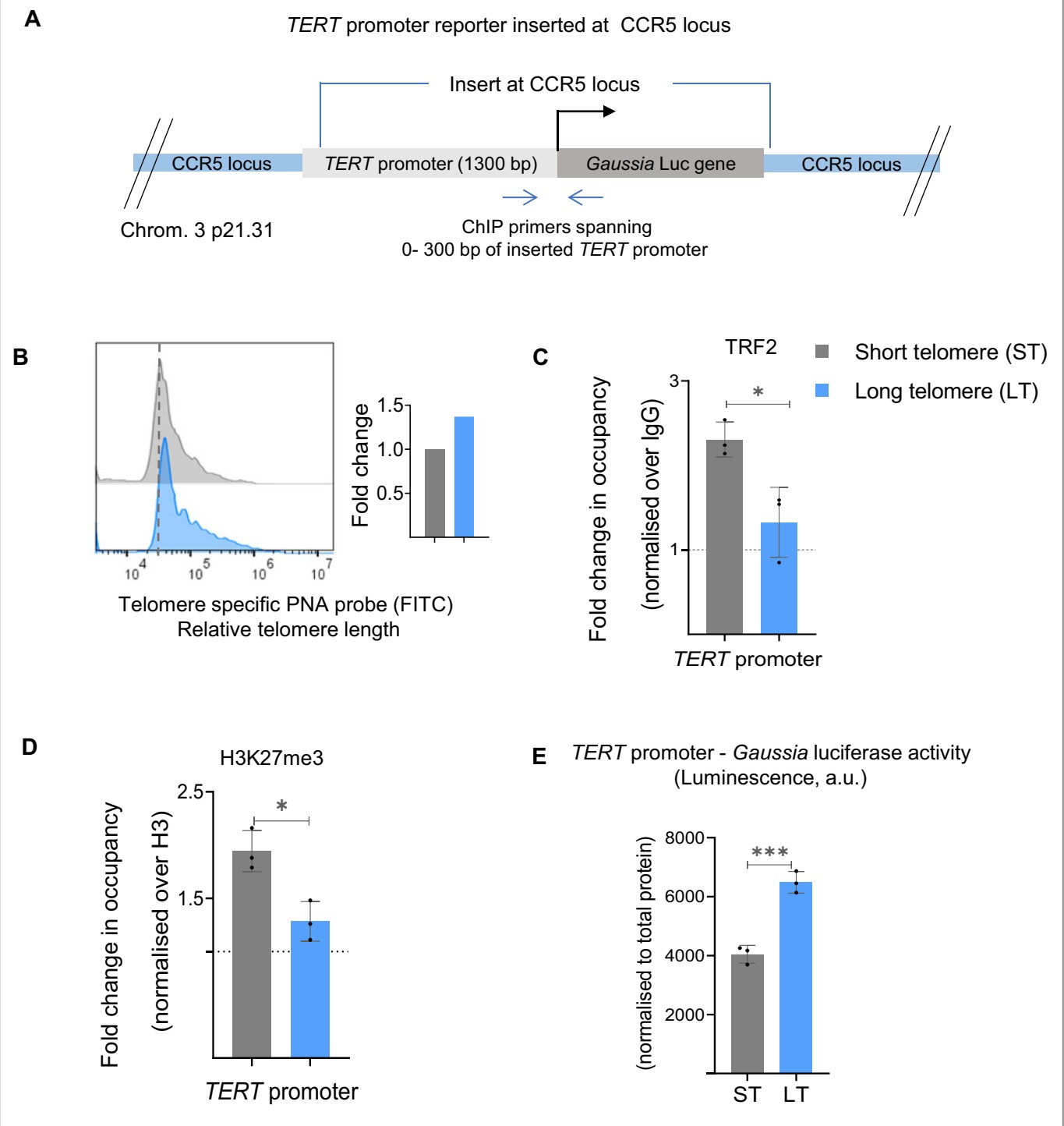

**Figure 4.** Artificially inserted telomerase reverse transcriptase (*TERT*) promoter shows telomere length–dependent regulation. (**A**) Scheme depicting CRISPR modified HEK293T cells with 1300 bp *TERT* promoter driving *Gaussia* luciferase (*Gaussia* Luc) construct inserted at the CCR5 safe harbour locus. Scheme denotes ChIP primers used to study chromatin occupancy of 0–300 bp *TERT* promoter region inserted at the exogenous locus. (**B**) Relative fold change in telomere length in *TERT* promoter insert cells following telomere shortening determined by FACS; quantification in the right panel. (**C,D**) ChIP followed by qRT-PCR at the 0–300 bp *TERT* promoter (upstream of TSS) insert at CCR5 locus for TRF2 (**C**), and H3K27me3 (**D**) in ST/LT cells. Occupancy normalised to respective IgG or total histone H3 (for H3K27me3). (**E**) *TERT* promoter-*Gaussia* luciferase activity in ST cells over LT cells from inserted exogenous with *TERT* promoter. Reporter activity is presented as luminescence (arbitrary units, a.u.) normalised to respective total protein levels. Error bars represent ± SDs from the mean of three independent biological replicates of each experiment. P-values are calculated by an unpaired t-test for all

*Figure 4 continued on next page*

*Figure 4 continued*

data except mRNA for MDA-MB-231-ST/LT and HCT116-ST/LT Telomere trimming-Cas9 models where paired t-test was performed. (*p<0.05, **p<0.01, ***p<0.005, ****p<0.0001).

The online version of this article includes the following source data for figure 4:

**Source data 1.** Source data for all plots in *Figure 4*.

and H3K27me3 occupancy were as expected from a decrease/increase in TRF2 binding at the *TERT* promoter as TL increased and then receded.

Consistent with this, expression of the endogenous *TERT* (using 3'UTR specific qRT-PCR primers) increased by Day 10 from Day 0 levels, and was significantly reduced by Day 24 compared to the Day 10 levels, although not reduced to the Day 0 state (*Figure 5G*), supporting the role of TL elongation in inducing or TL shortening in suppressing *TERT*.

We considered a second temporal model using stable MDA-MB-231 cells with dox-inducible *TERT* to exclude the possibility of cell-type-specific effects. After dox induction, we followed TL elongation/shortening and telomerase activity (through days 4/8/10 and 14, *Figure 6—figure supplement 1A–B*, ++/-- denotes presence/absence of dox, respectively). Like HT1080 cells, here also TL elongated till Day 10 and beyond Day 10, the presence of dox showed no further elongation in TL (data not shown). Like the HT1080 cell model, dox was withdrawn at Day 10, however, in the case of MDA-MB-231 cells, TL shortened to roughly the initial (Day 0) state by Day 14, in a relatively short timeframe than what was noted for HT1080 cells (*Figure 6—figure supplement 1A*).

For detailed analysis, here we selected days 0 and 10 with dox, withdrawal of dox at Day 10 (consistent with the HT1080 cell model), and thereafter Day 14 (given the enhanced reduction in TL in MDA-MB-231 compared to HT1080 cells) as representative time points (Dox-MDA-MB-231; *Figure 6A*). TL elongated to ~twofold by Day 10; and receded to approximately the initial (Day 0) length by Day 14 after stopping dox at Day 10 (*Figure 6B*).

Decrease in TRF2 binding, and corresponding reduced REST, EZH2, and the H3K27me3 mark at the *TERT* promoter were clear on Day 10 relative to Day 0 (*Figure 6C–F*). Promoter TRF2, REST, and EZH2 occupancy, including H3K27 trimethylation, was regained in the subsequent time point of Day 14, relative to the Day 10 levels, as expected from TL shortening after Day 10 (*Figure 6C–F*). Accordingly, we noted endogenous *TERT* transcript followed TL elongation/shortening: increased progressively till Day 10, and subsequently reduced gradually at Day 14 (*Figure 6G*). Cell-type specific differences in the experimental models, for instance, relatively low regain of TRF2 occupancy at the final time day points, Day 14 in MDA-MB-231 compared to Day 24 HT1080 cells, other than relatively enhanced pace of shortening in MDA-MB-231 cells vis-à-vis HT1080 cells on dox withdrawal were interesting to note. Nevertheless, taken together, data from two independent models of temporally altered TL, further supported promoter TRF2-binding mediated *TERT* regulation to be TL-dependent.

## Telomere-dependent *TERT* regulation operates in vivo within xenograft tumours

We next sought to test TL-sensitive TRF2 binding at the *TERT* promoter in vivo. Xenograft tumours were grown with either HT1080-ST or HT1080-LT cells in NOD/SCID mice (n=5 each, for tumour size characterization see Methods, *Figure 7A*). Following harvesting, tumours were first characterized: TL was overall higher in all LT tumours compared to HT1080-ST tumours; increased *TERT* and telomerase activity was retained in HT1080-LT tumours relative to ST (*Figure 7B and C*; *Figure 7—figure supplement 1A*). TRF2-ChIP from tumour tissue showed HT1080-ST had significantly more TRF2 occupancy at the *TERT* promoter than HT1080-LT, consistent with results from cultured cells (*Figures 7D and 2B*). Correspondingly, H3K27me3 mark deposition was reduced in xenograft HT1080-LT tumours compared to HT1080-ST (*Figure 7E*), supporting that TL-dependent *TERT* regulation was retained in cells growing in vivo.

## G-quadruplex-mediated TRF2 binding is essential for telomere-dependent *TERT* regulation

G-quadruplex (G4) DNA secondary structures (*Bryan and Baumann, 2011*), widely reported as gene regulatory motifs (*Rawal et al., 2006*; *Huppert and Balasubramanian, 2007*; *Verma et al.,*

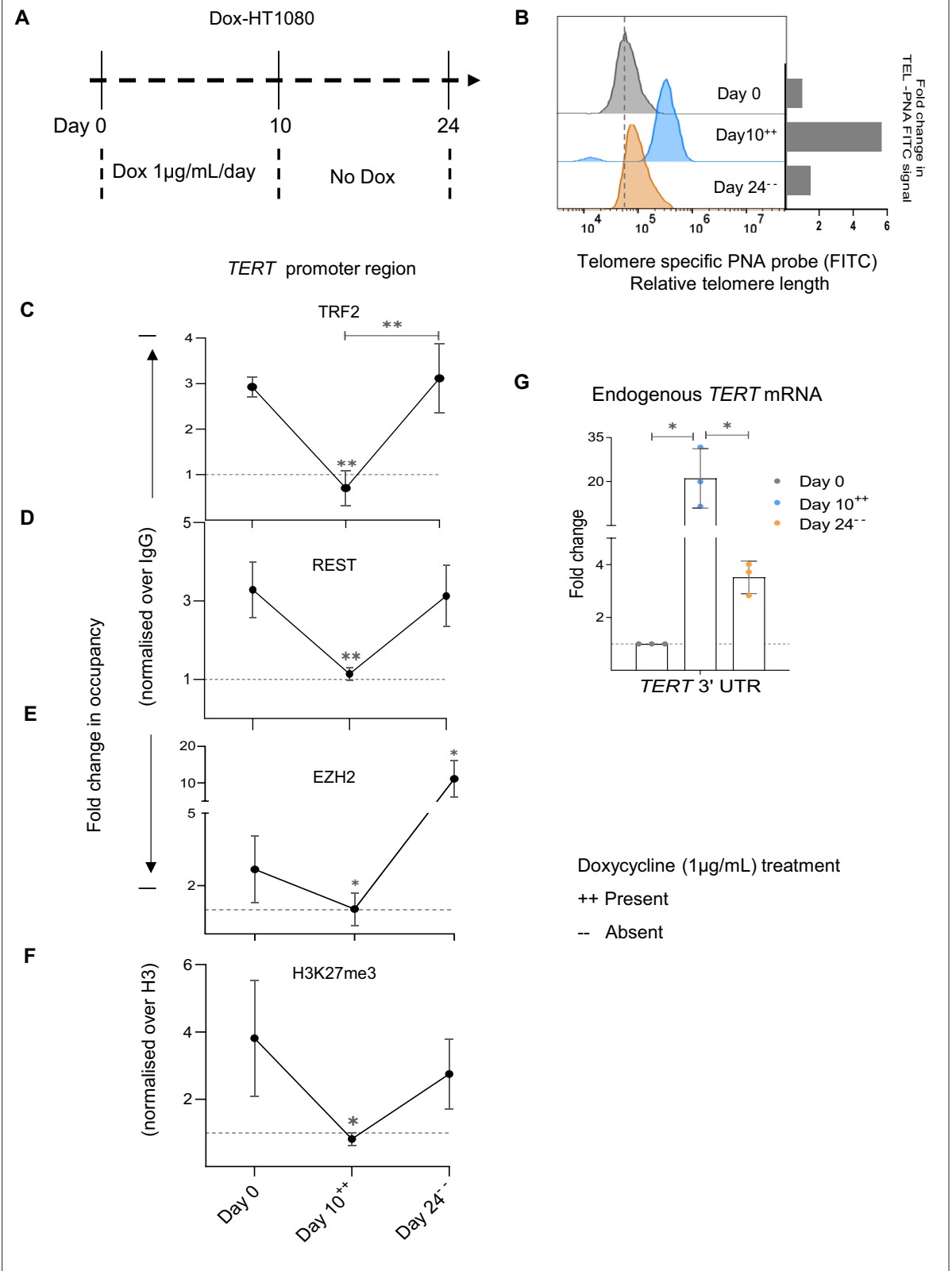

**Figure 5.** Temporal telomere length elongation followed by shortening shows telomere-sensitive transcriptional regulation of telomerase reverse transcriptase (*TERT)* in HT1080 cells. (**A**) Scheme depicting the protocol followed for doxycycline (Dox) inducible TERT overexpression in HT1080 cells (Dox-HT1080). ++/-- denotes the presence/ absence of dox at the indicated day points. (**B**) Relative fold change in telomere length at Day 0,10, and 24 determined by FACS in Dox-HT1080 cells; quantification in right panel. (**C–F**) ChIP followed by qRT-PCR at the 0–300 bp *TERT* promoter (upstream of

*Figure 5 continued on next page*

*Figure 5 continued*

TSS) for TRF2 (**C**) REST (**D**) EZH2 (**E**) or H3K27me3 (**F**) in Dox-HT1080 cells at Day 0,10, and 24; occupancy normalised to respective IgG or total Histone H3 (for H3K27me3). (**G**) *TERT* mRNA expression by qRT-PCR using *TERT*-specific 3'UTR primers in Dox- HT1080 cells at day intervals (as indicated); normalised to *GAPDH* mRNA levels. Fold changes were calculated independently for each biological replicate, as the three conditions represent paired samples (uninduced, induced with doxycycline, and post-doxycycline withdrawal). Error bars represent ± SDs from the mean of three independent biological replicates of each experiment. One-way ANOVA followed by post-hoc tests (Tukey's HSD) was performed to compare means across time points in Figs C-G (*p<0.05, **p<0.01, ***p<0.005, ****p<0.0001).

The online version of this article includes the following source data and figure supplement(s) for figure 5:

**Source data 1.** Source data for all plots in *Figure 5*.

**Figure supplement 1.** Temporal telomere length elongation followed by shortening shows telomere-sensitive transcriptional regulation of telomerase reverse transcriptase (*TERT)* in HT1080 cells (continued).

**Figure supplement 1—source data 1.** Source data for all plots in *Figure 5—figure supplement 1*.

*2008*; *Verma et al., 2009*), were found in the *TERT* promoter (*Palumbo et al., 2009*; *Lim et al., 2010*). G4-dependent promoter TRF2 binding was also recently shown to regulate *TERT* (*Sharma et al., 2021*). Here, we asked, whether and how TL-dependent *TERT* regulation was affected by the promoter G4s. We used the *TERT* promoter-*Gaussia* Luc-reporter inserted at the CCR5 locus in HEK293T cells described above (*Figure 4A*). G>A mutations at –124 or –146 positions from the TSS of the *TERT* promoter are frequently found to be clinically associated with multiple cancers and reported to disrupt the G4s (*Akincilar et al., 2016b*; *Huang et al., 2013*; *Horn et al., 2013*; *Killela et al., 2013*; *Kang et al., 2016*; *Li et al., 2016*; *Akincilar et al., 2016a*). These mutations were introduced in the reporter individually and a pair of cell lines with either long or short TL were generated (*Figure 8A–B*).

Low promoter TRF2 binding and reduced H3K27 trimethylation in cells with either –124 or –146 G>A mutations within the *TERT* promoter-reporter inserted at the CCR5 locus was clear from earlier work (*Sharma et al., 2021*). Here, based on low TRF2 binding at the G4-disrupted *TERT* promoter, we reasoned that TL-dependence of *TERT* regulation would be affected. TRF2 binding at the *TERT* promoter with G4-disrupting mutations was not regained upon shortening of TL and was consistent in the case of both –124G>A and –146 G>A mutations (*Figure 8C*, left and right panels). This was in contrast to the increase in promoter TRF2 binding observed on TL shortening in the case of the unmutated promoter (*Figure 4C*), supporting the role of G4s in *TERT* promoter TRF2 binding and thereby in TL-dependent *TERT* regulation. As expected, the difference in H3K27me3 deposition and *Gaussia* Luc activity was insignificant in –124G>A or –146 G>A ST cells compared to the corresponding LT cells (*Figure 8D and E*).

TRF2 R17 residue is required for the repression of *TERT* expression

To understand the role of TRF2 post-translational modification(s) (PTM), if any, on *TERT* repression, we screened TRF2 mutants (*Walker and Zhu, 2012*). TRF2 mutants arginine methylation-deficient R17H (*Mitchell et al., 2009*) acetylation-deficient K176R, K190R (*Rizzo et al., 2017*), or phosphorylation-deficient T188N (*Tanaka et al., 2005*) were expressed in HT1080 cells in a background where endogenous TRF2 was silenced (*Figure 9—figure supplement 1A*). As expected, TRF2 silencing induced *TERT*, and re-expression of TRF2-wild type (WT) repressed *TERT*. Interestingly, R17H and K176R increased *TERT*, whereas T188N and K190R gave relatively moderate change (*Figure 9— figure supplement 1B*). Noting that the N-terminal domain of TRF2 (with the R17 residue) was shown to interact with G4 DNA (*Bryan and Baumann, 2011*; *Pedroso et al., 2009*), and also the core histone H3 (*Konishi et al., 2016*), we focused on understanding the role of TRF2-R17 at the *TERT* promoter.

Dox-inducible TRF2-R17H or TRF2-WT stable lines were made with HT1080, HCT116, and MDA-MB-231 cells. Induction of TRF2-R17H gave significant upregulation of *TERT*; whereas, as expected, TRF2-WT repressed *TERT* in all three cell lines (*Figure 9A*, *Figure 9—figure supplement 1C*); dox dose-dependent TRF2 induction and resultant *TERT* levels shown in *Figure 9—figure supplement 1D*. The increase in *TERT* with TRF2-R17H expression in contrast to its reduction with TRF2-WT, was also clear from *TERT* FISH (*Figure 9B*).

To understand loss of function, we looked closely at TRF2-R17H DNA binding at the *TERT* promoter. The binding of FLAG-tagged-TRF2-R17H on the *TERT* promoter was insignificant compared to the FLAG tagged-TRF2-WT in HT1080 cells (*Figure 9C*). As expected from the loss of TRF2-R17H binding, REST, EZH2 and H3K27me3 occupancy was significantly low on the *TERT* promoter for TRF2-R17H compared to TRF2-WT (*Figure 9D–F*).

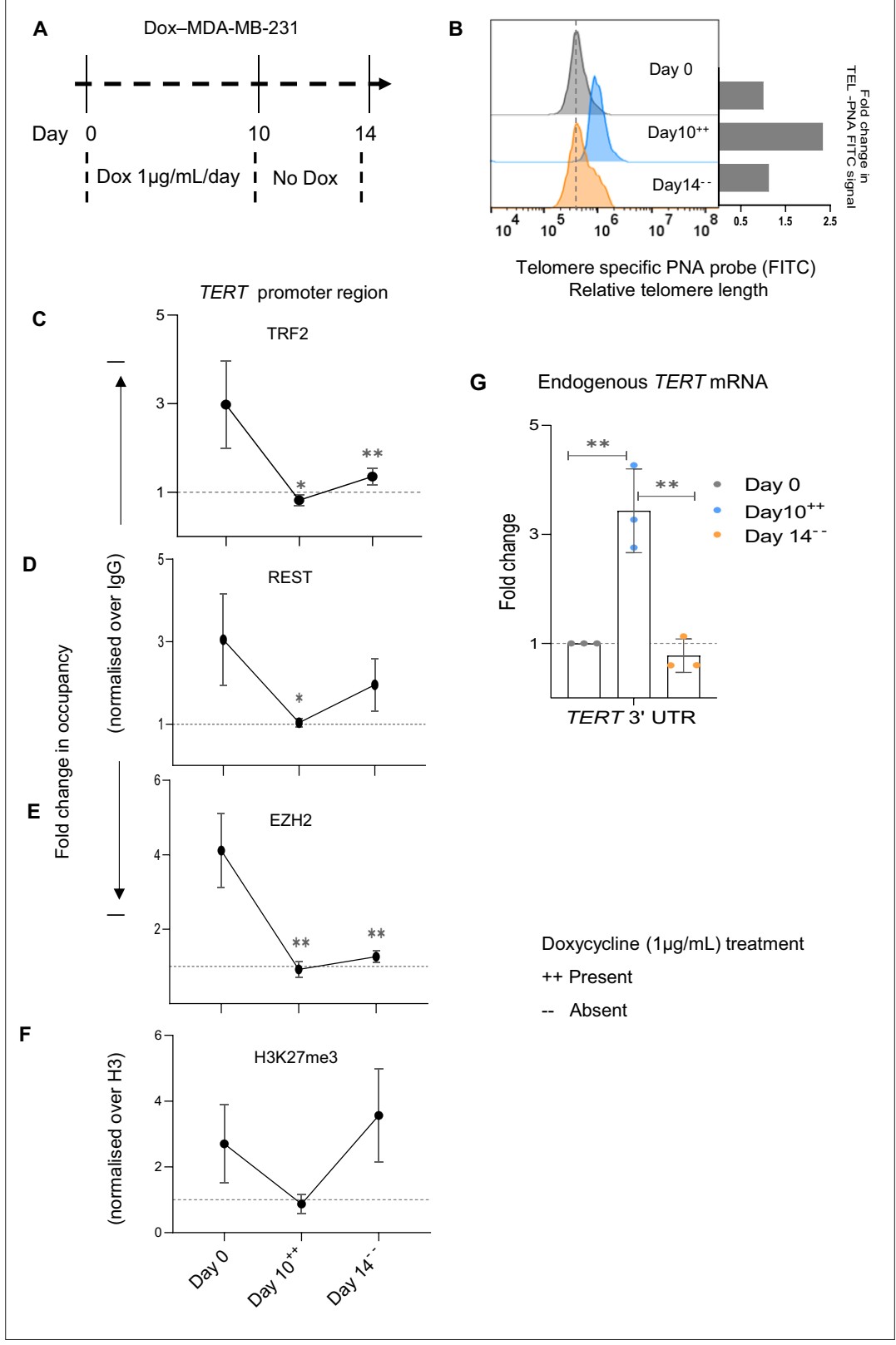

**Figure 6.** Temporal telomere length elongation followed by shortening shows telomere-sensitive transcriptional regulation of telomerase reverse transcriptase (*TERT*) in MDAMB 231 cells. (**A**) Scheme depicting protocol followed for doxycycline (Dox) inducible TERT overexpression in MDA-MB-231 (Dox-MDA-MB-231). ++/-- denotes the presence/ absence of dox at the indicated day points. (**B**) Relative fold change in telomere length at Day 0,10,

*Figure 6 continued on next page*

*Figure 6 continued*

and 14 determined by Flow cytometry in Dox-MDA-MB-231; quantification in right panel. (**C–F**) ChIP followed by qRT-PCR at the 0–300 bp *TERT* promoter (upstream of TSS) for TRF2 (**C**) REST (**D**) EZH2 (**E**) or H3K27me3 (**F**) in Dox-MDA-MB-231 cells at Day 0,10, and 14; occupancy normalised to respective IgG or total Histone H3 (for H3K27me3). (**G**) *TERT* mRNA expression by qRT-PCR using TERT-specific 3'UTR primer in Dox- MDA-MB-231 cells at day intervals (as indicated); normalised to *GAPDH* mRNA levels. Fold changes were calculated independently for each biological replicate, as the three conditions represent paired samples (uninduced, induced with doxycycline, and post-doxycycline withdrawal). Error bars represent ± SDs from the mean of three independent biological replicates of each experiment. One-way ANOVA followed by post-hoc tests (Tukey's HSD) was performed to compare means across time points in Figs C-G (*p<0.05, **p<0.01, ***p<0.005, ****p<0.0001).

The online version of this article includes the following source data and figure supplement(s) for figure 6:

**Source data 1.** Source data for all plots in *Figure 6*.

**Figure supplement 1.** Temporal telomere length elongation followed by shortening shows telomere-sensitive transcriptional regulation of telomerase reverse transcriptase (*TERT)* in MDAMB 231 cells (continued).

**Figure supplement 1—source data 1.** Source data for all plots in *Figure 6—figure supplement 1*.

## TRF2 promotes H3K27 trimethylation in vitro

For a deeper understanding of the role of TRF2 in H3K27 trimethylation, we used in vitro histone H3 methyltransferase assay. Purified histone H3 along with the reconstituted PRC2 repressor complex was analyzed in the presence/absence of recombinant TRF2-WT or TRF2-R17H for methyltransferase activity (*Figure 9G*, *Figure 9—figure supplement 1E-F*). H3K27 trimethylation in the presence of H3 and the PRC2 complex was first confirmed. Following this, treatment with purified TRF2-WT gave further increase in H3K27 trimethylation, whereas TRF2-R17H did not lead to any significant increase, relative to respective controls (*Figure 9G*). Interestingly, together these support a direct function of TRF2 in histone H3K27 trimethylation in presence of the PRC2 complex, where the R17 residue of TRF2 plays a necessary role.

## Telomere length regulates *TERT* expression in induced pluripotent stem cells (iPSCs)

To test the effect of TL on *TERT* expression in a physiological setting where telomeres elongate or shorten, we used fibroblast cells along with corresponding iPSCs. Primary foreskin (FS) fibroblasts were reprogrammed to pluripotent stem cells (iPSCs) (see Methods, scheme in *Figure 10A*, characterization data for iPSC in *Figure 10B*). *TERT* mRNA expression (*Figure 10C*), telomerase activity (*Figure 10D*) and relative telomere length (*Figure 10E*), were confirmed, and as expected were higher in the iPSCs compared to parent FS fibroblast cells. Chromatin binding of TRF2, REST, and EZH2 proteins and deposition of H3K27me3 mark were analysed in FS fibroblast and derived iPSCs (*Figure 10F–I*). TRF2 occupancy in the iPSCs with longer telomeres was found to be lower on the *TERT* promoter in comparison to parent fibroblasts (*Figure 10F*). Furthermore, occupancy of REST (*Figure 10G*) and the repressor complex protein, EZH2 (*Figure 10H*) was reduced in the pluripotent stem cells. Corresponding to the reduced levels of TRF2 occupancy and resultant lower binding of the REST/PRC2 complex on the *TERT* promoter in the iPSCs, the repressor mark, H3K27me3 was significantly lower relative to FS Fibroblast cells (*Figure 10I*).

Next, we asked if TL played a causal role in *TERT* expression. That is, whether TL change in iPSCs influenced TRF2 occupancy at the *TERT* promoter and its subsequent impact on *TERT*. We used the telomere-specific sgRNA-guided CRISPR-Cas9 to trim telomeres (as reported earlier) (*Chen et al., 2013*). This resulted in a population of iPSCs with short telomeres ('iPSC-ST';~50% reduction in TL; see Methods, *Figure 11A and B*). A comparison of TRF2 occupancy between iPSC-ST cells and unaltered iPSCs revealed a marked increase in TRF2 binding at the *TERT* promoter (~ threefold) in the telomere-shortened cells (*Figure 11C*). As expected from this, based on the above findings, we observed an increase (of 1.5-fold) in the deposition of the repressive histone mark H3K27me3 in iPSC-ST cells compared to unaltered iPSCs (*Figure 11D*). Further, consistent with these, *TERT* was significantly reduced in iPSC-ST cells relative to unaltered iPSCs (*Figure 11E*). Taken together, these demonstrate how *TERT* repression results of TL shortening through non-telomeric TRF2.

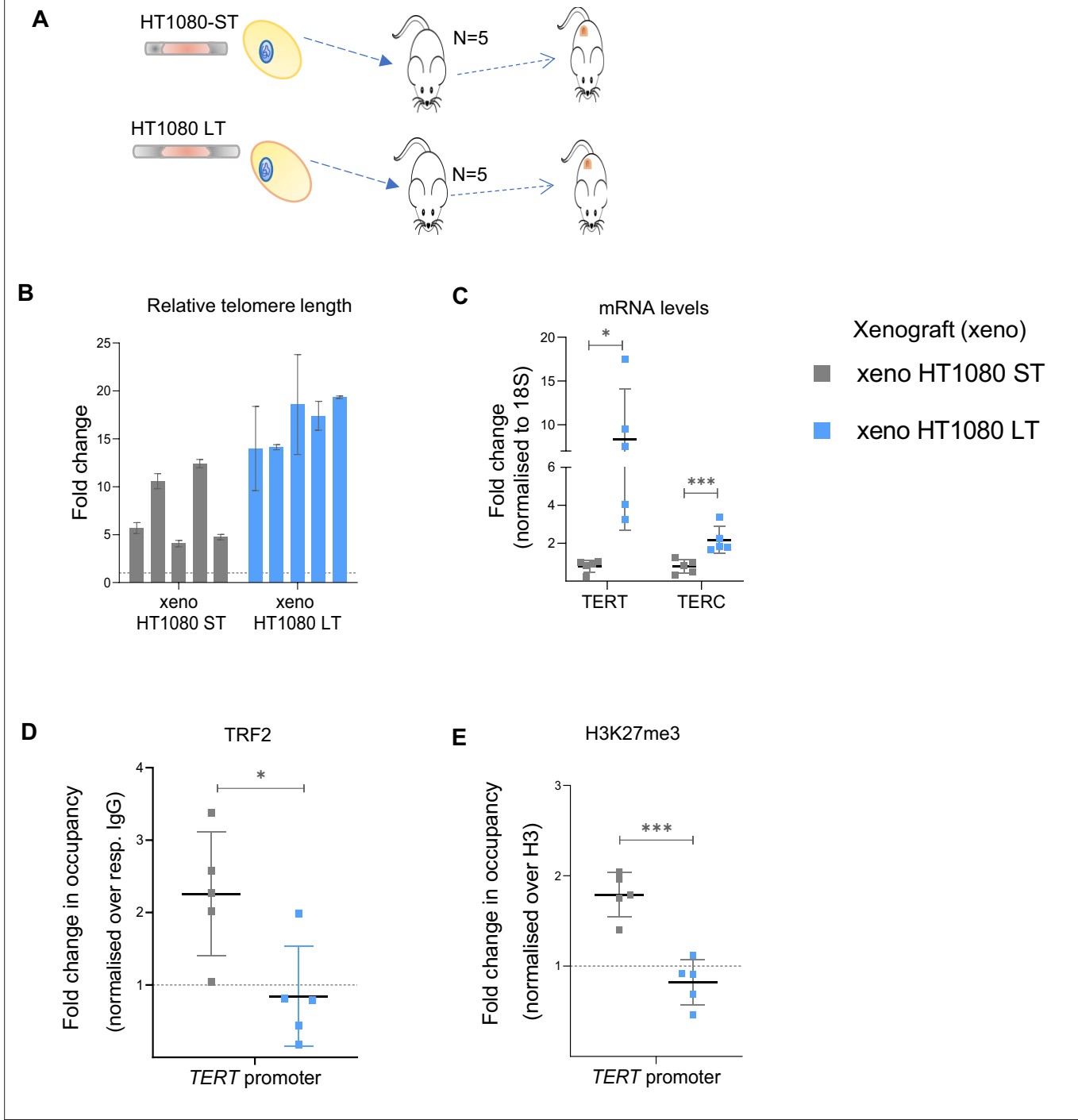

**Figure 7.** Telomere length-sensitive telomerase reverse transcriptase (*TERT*) regulation in in vivo tumour xenografts (**A**) Scheme depicting the generation of tumour xenograft with HT1080-ST or HT1080-LT cells. (**B**) Relative fold change in telomere length in xenograft samples determined by qRT-PCR-based telomere length detection method as reported earlier (*O'Callaghan et al., 2008* and *Cawthon, 2002*). Telomeric signal normalised over single copy gene, 36B4. (**C**) *TERT* (exon 15/16 full-length transcript) and *hTERC* mRNA expression in xenograft tissues by qRT-PCR; normalised to 18 S mRNA levels. (**D, E**) ChIP followed by qPCR at the 0–300 bp *TERT* promoter (upstream of TSS) for TRF2 (**D**) and H3K27me3 (**E**); occupancy normalised to respective IgG and total H3 (for H3K27me3). Error bars represent ± SDs across individual values of n=5 xenograft tumour samples in each group. P-values are calculated by unpaired t-test with Welch's correction (*p<0.05, **p<0.01, ***p<0.005, ****p<0.0001).

The online version of this article includes the following source data and figure supplement(s) for figure 7:

**Source data 1.** Source data for all plots in *Figure 7*.

*Figure 7 continued on next page*

*Figure 7 continued*

**Figure supplement 1.** Telomere length-sensitive telomerase reverse transcriptase (*TERT*) regulation in in vivo tumour xenografts (continued).

**Figure supplement 1—source data 1.** Source data for all plots in *Figure 7—figure supplement 1*.

## Discussion

Relatively long telomeres induced permissive chromatin at the *TERT* promoter. Conversely, shorter telomeres led to closed chromatin at the promoter (*Figures 3A–B and 5D–E*, *Figure 6D–E*). This resulted in telomere-dependent *TERT* transcription in multiple cell types/models as in: (a) short/long telomere HT1080, HCT116, MDA-MB-231 cells, and primary FS fibroblast along with corresponding derived iPSC; (b) temporal TL elongation with subsequent shortening; (c) TL-dependent transcription from artificially inserted *TERT* promoter; (d) and, tumour cells grown in vivo in mice. Mechanistically, TRF2 binding at the *TERT* promoter decreased/increased with TL elongation/shortening, respectively, affecting TRF2-dependent recruitment of epigenetic modulators REST and the PRC2 repressor complex (*Figures 2–11*). The resulting loss/gain in the repressor histone H3K27 trimethylation up or down-regulated *TERT,* respectively, as is shown in the following scheme (*Figure 12*). This was consistent with the TSP model described by us earlier (*Figure 1*; *Sharma and Chowdhury, 2022*; *Mukherjee et al., 2018b*; *Vinayagamurthy et al., 2020*).

The overwhelming understanding, particularly in cancer cells, suggests that the reactivation of *TERT* drives telomere elongation/maintenance (*Shay and Wright, 2019*; *Artandi and DePinho, 2010*; *Roake and Artandi, 2020*; *Smogorzewska and de Lange, 2004*). On the other hand, however, the possibility that telomeres conversely might impact the regulation of *TERT* remains unclear. To test this, here we focused on two contexts: cancer cell lines engineered to modify TL and cellular reprogramming where TL changes are evident (*Flores et al., 2008*; *Allsopp, 2012*). Multiple lines of evidence support the causal role of telomeres in *TERT* regulation. (a) TL elongation, by exogenous *TERT*, or independent of *TERT* (using G-rich telomeric oligonucleotides as reported earlier (see Methods)) in different cell types produced *TERT* upregulation (*Figures 2–4*). (b) TL shortening, on the other hand, caused *TERT* suppression (*Figures 2–4*). (c) Temporally induced TL elongation over days, using exogenous *TERT*, gradually activated the endogenous *TERT* promoter. As TL receded, after discontinuing induction, the endogenous *TERT* promoter was re-suppressed (*Figures 5 and 6*). (d) *TERT* activation during reprogramming of primary fibroblast to iPSCs showed reduced promoter TRF2 binding on telomere elongation and *TERT* activation (*Figure 10*). (e) And, finally upon telomere shortening in iPSCs, TRF2 occupancy was regained on the *TERT* promoter, followed by increased repressor mark deposition and reduced *TERT* transcription (*Figure 11*). Together, these demonstrate that telomeres play a key role in *TERT* regulation, adding new insight to the current understanding of telomere elongation/maintenance as a consequence of *TERT* activity (*Artandi and DePinho, 2010*; *Jafri et al., 2016*; *Giardini et al., 2014*).

Telomere length and *TERT* expression are critical in maintaining pluripotency and the ability of stem cells to differentiate (*Flores et al., 2008*; *Allsopp, 2012*; *Armstrong et al., 2005*; *Yang et al., 2008*). The re-occupation of TRF2 at the *TERT* promoter in telomere-shortened iPSCs (iPSC-ST) along with suppression of its expression demonstrates the applicability of TRF2-dependent TL-sensitive *TERT* regulation in stem cells and presents a potentially useful tool to modulate its differentiation capacity and pluripotency with further exploration.

An earlier paper showed long telomeres on chromosome 5 p interacted with the proximal *TERT* locus (1.2 Mb away) by physical chromatin looping (TPE-OLD) (*Kim et al., 2016*). Resulting heterochromatinisation due to telomeric factors, including TRF2, repressed *TERT*; loss of looping on telomere shortening de-repressed *TERT*. We, on the other hand, observed that long telomeres induce, and shorter telomeres repress *TERT* transcription, suggesting this was distinct from looping-induced interactions. An important underlying difference in the nature of TRF2 at the *TERT* promoter must be noted: in TPE-OLD, TRF2 is telomere-associated, whereas TRF2 is non-telomeric in TSP. To confirm non-telomeric TRF2 binding, we reasoned in two ways. (a) Telomere-associated TRF2 is likely to be present with other telomeric factors like RAP1 and TRF1(21), and (b) inserted a *TERT* promoter-reporter >40 Mb from telomeres where looping interaction was unlikely. While TRF2 binding at the *TERT* promoter was clear, RAP1 or TRF1 was absent (*Sharma et al., 2021*). Furthermore, *TERT* promoter activity from the inserted reporter was TL-dependent (*Figure 4*). These support *TERT* promoter-bound TRF2 as

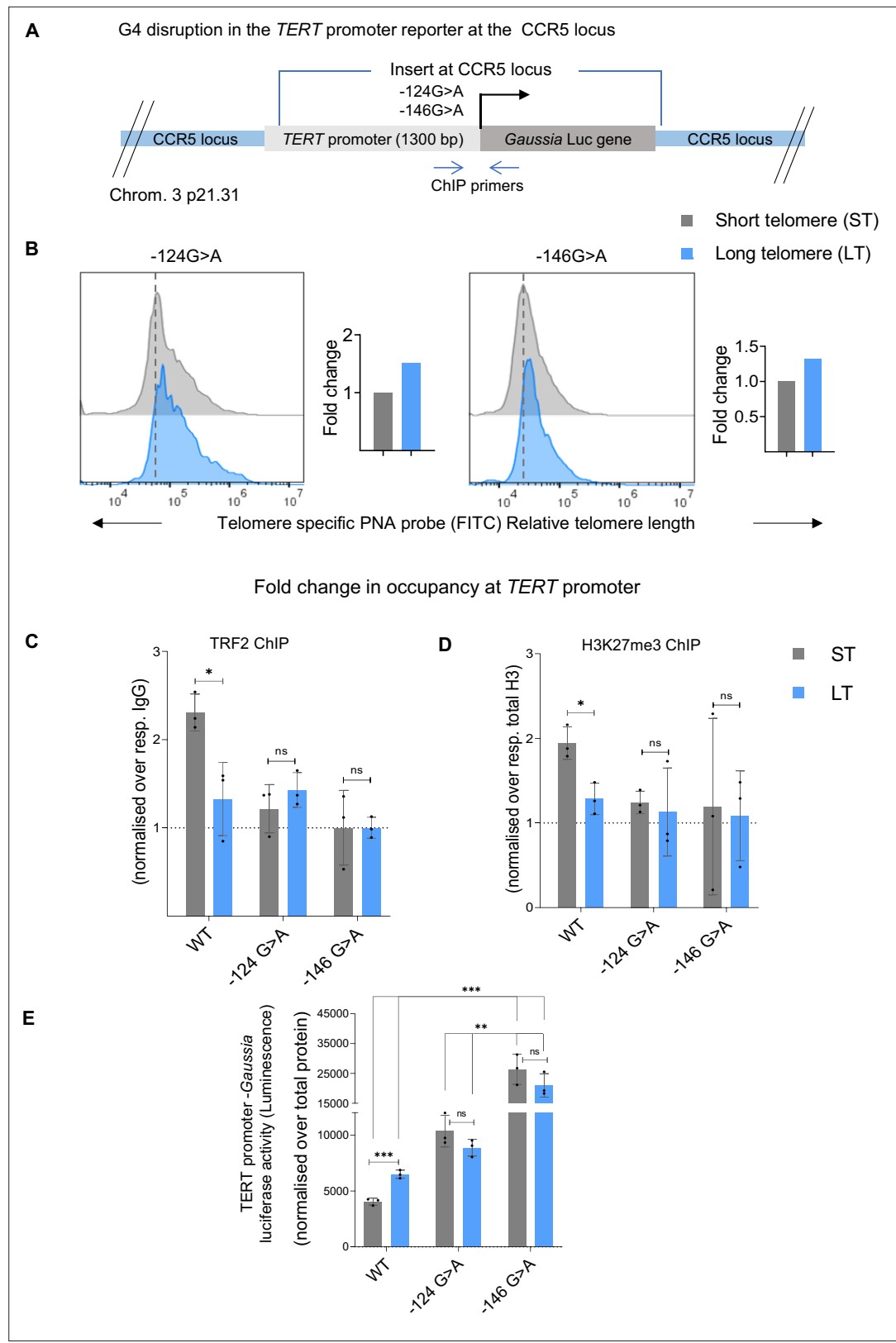

**Figure 8.** G-quadruplex-mediated TRF2 binding is essential for telomere-dependent telomerase reverse transcriptase (*TERT*) regulation. (**A**) Scheme depicting CRISPR-modified HEK293T cells with 1300 bp *TERT* promoter with G>A substitution at –124 or –146 bp (upstream of TSS) driving *Gaussia* luciferase (*Gaussia* Luc) construct at the CCR5 safe harbour locus. Scheme denotes ChIP primers used to study chromatin occupancy of

*Figure 8 continued*

0–300 bp *TERT* promoter region inserted at the exogenous locus. (**B**) Relative fold change in telomere length in two independent pairs of cells with short or long telomeres containing *TERT* promoter G4 disrupting mutations (–124G>A or –146 G>A) at CCR5 locus *TERT* promoter insert as determined by Flow cytometry; quantification in right panel. (**C**, **D**) ChIP followed by qRT-PCR at the inserted *TERT* promoter (0–300 bp upstream of TSS) for TRF2 (**C**), and H3K27me3 (**D**) in pairs of cells generated with –124G>A or –146 G>A mutation with long/ short telomeres, along with WT promoter ST/LT pair (as in *Figure 4C and D*). Occupancy normalised to respective IgG or total histone H3 (for H3K27me3). (**E**) *TERT* promoter-*Gaussia* luciferase activity in short or long telomere cells with - 124G>A and –146 G>A mutated *TERT* promoter sequence, along with WT promoter ST/LT pair (as in *Figure 4E*). Reporter activity presented as luminescence (arbitrary units, a.u.) normalised to respective total protein levels. Error bars represent ± SDs from the mean of three independent biological replicates of each experiment. Statistical significance was determined by two-way ANOVA followed by Tukey's post hoc test for all pairwise comparisons. For planned comparisons between each parental and short-telomere cell line, unpaired t-tests were used. (*$p<0.05$, **$p<0.01$, ***$p<0.005$, ****$p<0.0001$).

The online version of this article includes the following source data for figure 8:

**Source data 1.** Source data for all plats in *Figure 8*.

non-telomeric and indicate the two models (TPE-OLD and TSP) are likely context-specific. However, additional experimentation would be required to ascertain the contexts or range of telomere length at which the two mechanisms could operate in isolation or concert – providing further insights into their physiological relevance.

*TERT* promoter G4s had a causal role in telomere-induced *TERT* regulation (*Figure 8*). Interestingly, in many cancers, activation of *TERT* was reported to be frequently due to *TERT* promoter mutations; subsequently noted to overlap with the *TERT* promoter G4-forming stretch (*Akincilar et al., 2016b*; *Sharma and Chowdhury, 2022*; *Palumbo et al., 2009*; *Lim et al., 2010*; *Huang et al., 2013*; *Horn et al., 2013*; *Killela et al., 2013*; *Li et al., 2016*; *Akincilar et al., 2016a*). We show for the two most common clinically significant *TERT* promoter mutations known to abrogate these G4 structures, the TRF2 binding site on the *TERT* promoter was also disrupted (*Figure 8C*; *Sharma et al., 2021*). The artificially introduced *TERT* promoter-reporter on G4 disruption lost its dependence on TL (*Figures 4 and 8*). These build support for a G4-dependent TL-driven mechanism of *TERT* regulation/maintenance (*Heaphy et al., 2011*; *Hoang and O'Sullivan, 2020*).

Xenograft mice tumours indicated TL-induced *TERT* activation might be active in vivo. However, results are primarily from cells engineered for TL elongation/shortening (*Figure 7*). While these open new avenues, additional work would be necessary to understand broader implications. For example, it is not clear how the initial TL elongation (which results in *TERT* reactivation) is triggered during tumorigenic transformation. Interestingly, recent work shows telomere transfer from antigen-presenting cells (APCs) results in telomere elongation of specific T cell populations, subsequent telomerase expression induced proliferative expansion during T cell activation (*Lanna et al., 2022*). This suggests the possibility of telomere transfer between tumour and immune/other cells within the tumour microenvironment might alter TL, contributing to *TERT* activation. A similar model might be functional during reprogramming. Moreover, it would be of interest to test how TL-dependent *TERT* regulatory mechanisms affect senescing primary cells and telomerase-negative cancer cells (with ALT mechanism).

The current study provides experimental evidence that TRF2, a well-characterized telomere-binding protein, mediates crosstalk between telomeres and the regulatory region of the *TERT* gene in a telomere length-dependent manner. Given the observed link between *TERT* expression and telomere length, it is likely that additional telomere-associated proteins and regulatory pathways contribute to this regulation.

The remaining shelterin complex components—POT1, hRap1, TRF1, TIN2, and TPP1—may play crucial roles in this context, as they are integral to telomere maintenance and protection (*Stewart et al., 2012*). Additionally, several DNA damage response (DDR) proteins, which interact with telomere-binding factors and help preserve telomere integrity, could potentially influence *TERT* regulation in a telomere length-dependent manner (*Longhese, 2008*). However, direct interactions or regulatory roles would require further experimental validation. Another group of proteins with potential relevance in this mechanism are the sirtuins, which directly associate with telomeres and are known

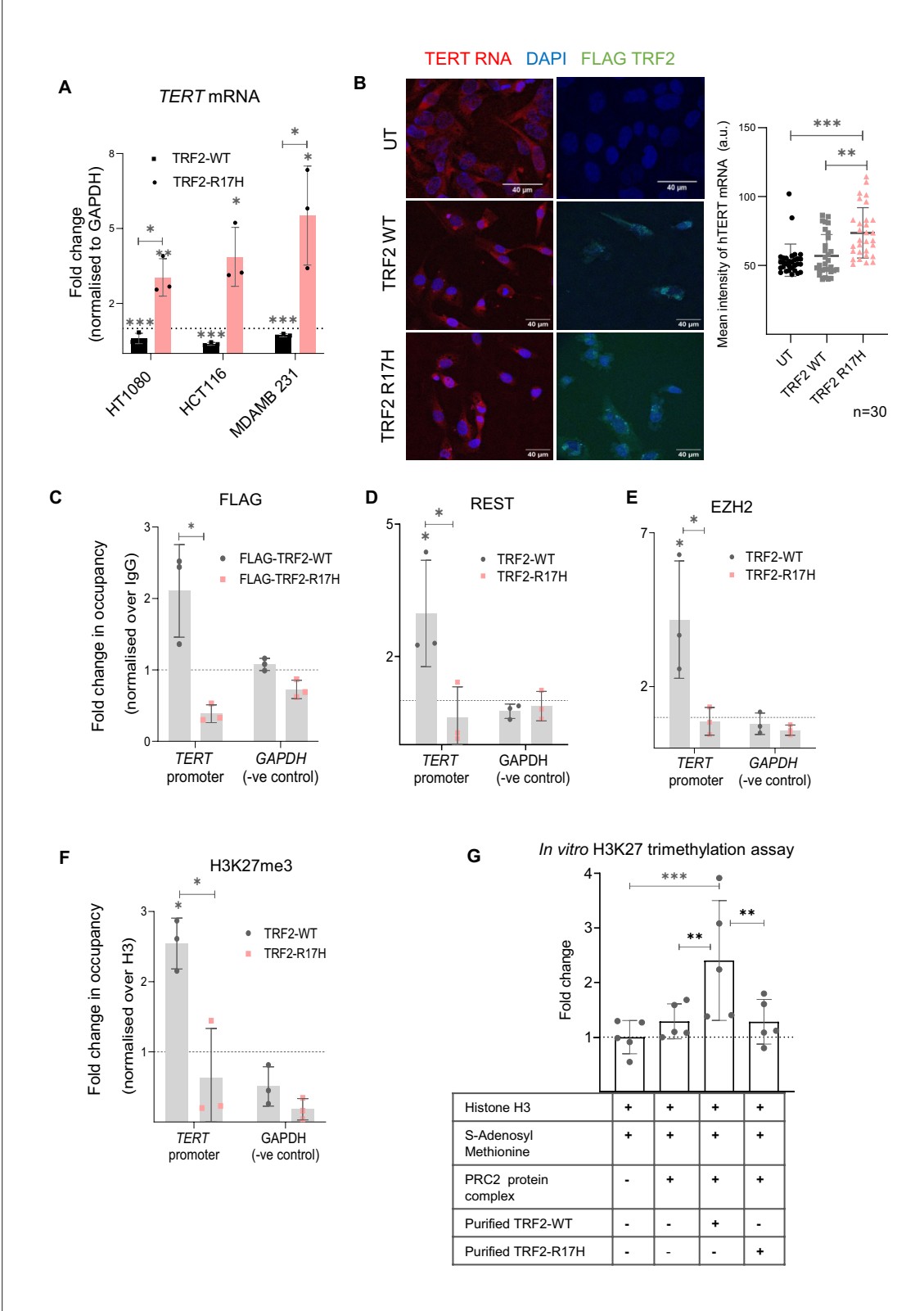

**Figure 9.** TRF2 R17 residue is required for the repression of telomerase reverse transcriptase (*TERT*) expression. (**A**) *TERT* full-length transcript (exon 15/16) mRNA expression levels by qRT-PCR; normalised to *GAPDH* mRNA levels upon stable doxycycline induction of wild-type (WT) TRF2 or R17H TRF2 mutants in HT1080, HCT116, or MDA-MB-231 cells. (**B**) *TERT* mRNA FISH upon TRF2 WT or TRF2 R17H overexpression (untransfected, UT as control) in HT1080 cells; quantification shown in right panel. FLAG-tagged TRF2 overexpression was confirmed by Immunofluorescence microscopy.

*Figure 9 continued on next page*

*Figure 9 continued*

(**C–F**) ChIP followed by qRT-PCR at the 0–300 bp *TERT* promoter (upstream of TSS) for FLAG-tagged TRF2 (**C**), REST (**D**), EZH2 (**E**), or H3K27me3 (**F**) in HT1080 cells upon expression of WT TRF2 or TRF2 R17H. Occupancy normalised to respective IgG and total Histone H3 (for H3K27me3); qRT-PCR on the *GAPDH* promoter was used as the negative control in all cases. (**G**) In vitro methyltransferase activity of the reconstituted PRC2 complex resulting in H3K27 trimethylation in the presence or absence of TRF2 WT or TRF2 R17H protein. Error bars represent ± SDs from the mean of three independent biological replicates of each experiment. Unpaired t-tests were conducted to assess the significance of each condition individually within the same cell line, and to compare the two conditions TRF2-WT or TRF2-R17H in (**A**); one-way ANOVA followed by post-hoc tests (Tukey's HSD) was performed to compare means across the three conditions in (**B**); P-values are calculated by unpaired t-test in (**C–F**); and two-way ANOVA followed by post-hoc tests (Tukey's HSD) in (**G**). (*$p<0.05$, **$p<0.01$, ***$p<0.005$, ****$p<0.0001$).

The online version of this article includes the following source data and figure supplement(s) for figure 9:

**Source data 1.** Source data of all plots in *Figure 9*, except for *Figure 9B*.

**Source data 2.** Source data of *TERT* mRNA FISH quantification in *Figure 9B*.

**Figure supplement 1.** TRF2 R17 residue is required for the repression of telomerase reverse transcriptase (*TERT*) expression(continued).

**Figure supplement 1—source data 1.** Source data of all plots in *Figure 9—figure supplement 1*.

**Figure supplement 1—source data 2.** PDF file containing original western blot for *Figure 9C* indicating the relevant bands and treatment.

**Figure supplement 1—source data 3.** Original image files for western blot for *Figure 9C*.

**Figure supplement 1—source data 4.** PDF file containing original western blot for *Figure 9E* indicating the relevant bands.

**Figure supplement 1—source data 5.** Original image files for western blot for *Figure 9E*.

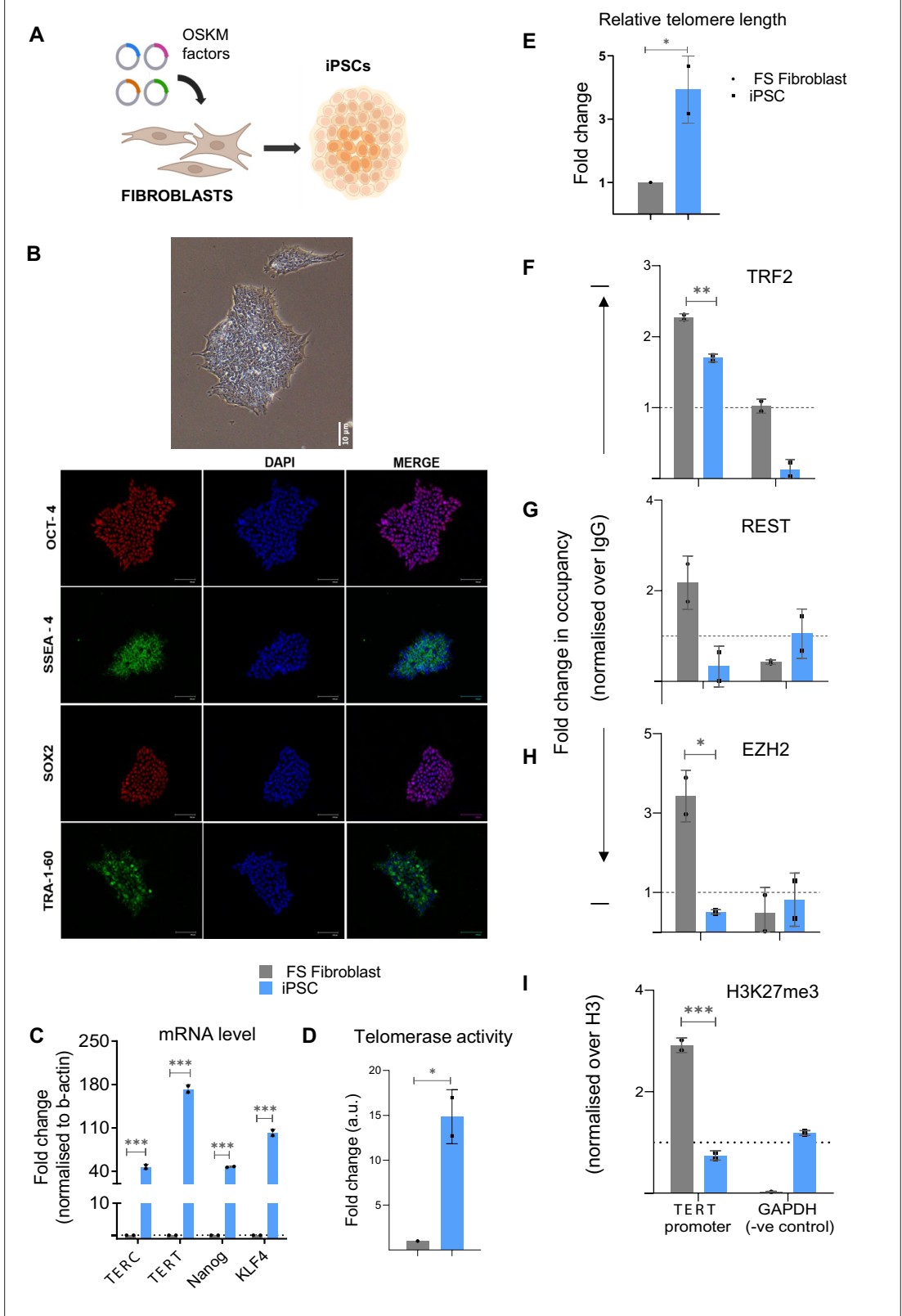

**Figure 10.** Telomere length influences telomerase reverse transcriptase (*TERT*) regulation in fibroblasts and derived iPSCs. (**A**) Scheme showing generation of induced pluripotent stem cells (iPSCs) from foreskin fibroblast (FS Fibroblast) cells by overexpressing Yamanaka factors (Oct4, Sox2, Klf4, Myc). (**B**) Characterization of iPSCs (Upper panel, bright field image) generated from FS Fibroblast cells by immunofluorescence using Oct-4, SSEA-4, Sox-2, and TRA-1–60 antibodies as stemness markers. (**C**) mRNA levels for *TERT* (full-length exon 15/16 transcript), *TERC* (RNA component),

*Figure 10 continued on next page*

*Figure 10 continued*

and stemness marker genes Nanog, Klf4 in FS fibroblast and derived iPSC, analysed in pairs in each biological replicates. (**D**) Telomerase activity in FS Fibroblast cells and derived iPSCs determined using telomerase-repeat-amplification-protocol (TRAP) followed by ELISA (see Methods). (**E**) Relative fold change in telomere length in primary FS Fibroblast cells and derived iPSC, determined by qPCR-based telomere length detection method. (**F–I**) ChIP followed by qPCR at the 0–300 bp *TERT* promoter (upstream of TSS) for TRF2 (**F**), REST (**G**), EZH2 (**H**), and H3K27me3 (**I**) up to 300 bp upstream of transcription start site (TSS); occupancy normalized to respective IgG or total Histone H3 (for H3K27me3). qPCR on the GAPDH promoter was used as a negative control in all cases. All error bars represent ± SDs from the mean of two independent biological replicates of each experiment. P-values are calculated by an unpaired t-test (*p<0.05, **p<0.01, ***p<0.005, ****p<0.0001).

The online version of this article includes the following source data for figure 10:

**Source data 1.** Source data of all plots in *Figure 10*.

**Source data 2.** Original images of iPSC IF in *Figure 10B*.

to positively regulate telomere length, undergoing repression upon telomere shortening (*Amano and Sahin, 2019b*; *Amano et al., 2019a*). Notably, SIRT1 has been reported to interact with telomerase (*Lee et al., 2024*), while SIRT6 has been implicated in TRF2 degradation (*Rizzo et al., 2017*) and telomerase activation (*Chen et al., 2021*). Given their roles in telomere homeostasis, sirtuins may serve as key mediators of telomere length-dependent *TERT* regulation.

In summary, results reveal a heretofore unknown mechanism of *TERT* regulation by TRF2 in a TL-dependent manner. Long telomeres sequester more TRF2 for telomere protection, which reduces TRF2 at the *TERT* promoter. Resulting in de-repression of *TERT*, and increased *TERT* levels in turn synthesize (*Figures 2–4*) and help maintain longer telomeres. The reversal of this was observed in TL shortening. Taken together, these suggest a feed-forward model mechanistically connecting telomeres to *TERT* with likely significant implications in better understanding telomere-related physiological events, particularly tumorigenesis, ageing, and de-differentiation (*Roake and Artandi, 2020*; *Vinayagamurthy et al., 2020*; *Jafri et al., 2016*; *Wong and Collins, 2003*; *Muñoz et al., 2006*; *Rossiello et al., 2022*).

# Materials and methods

## Key resources table

| Reagent type (species) or resource | Designation | Source or reference | Identifiers | Additional information |
|---|---|---|---|---|
| Cell Line (*Homo sapiens*) | HT0180 | ATCC | ATCC -CCL-121 RRID:CVCL_0317 | Fibrosarcoma |
| Cell Line (*Homo sapiens*) | MDAMB231 | ATCC | ATCC HTB-26 RRID:CVCL_0062 | Breast cancer |
| Cell Line (*Homo sapiens*) | HCT116 | ATCC | ATCC-CCL-247 RRID:CVCL_0291 | Colorectal carcinoma |
| Cell Line (*Homo sapiens*) | HEK293T | NCCS Cell Repository | RRID:CVCL_0063 | Embryonic kidney derived |
| Antibody | TRF2 (rabbit polyclonal) | Novus | (Novus NB110-57130) RRID:AB_844199 | ChIP (1:100), IP (1:100), WB (1:1000) |
| Antibody | REST | Millipore | #17–641 RRID:AB_1977463 | ChIP (1:100) |
| Antibody | Histone H3 (rabbit polyclonal) | Abcam | (Abcam ab1791)-RRID:AB_302613 | ChIP (1:100) |
| Antibody | GAPDH (mouse monoclonal) | Santa-cruz | # sc-32233, RRID:AB_627679 | WB (1:1000) |
| Antibody | Anti-Rabbit IgG (rabbit polyclonal) | Millipore | (Millipore 12–370) RRID:AB_145841 | isotype control (1:100) |

*Continued on next page*

*Continued*

| Reagent type (species) or resource | Designation | Source or reference | Identifiers | Additional information |
|---|---|---|---|---|
| Antibody | Anti-mouse IgG (mouse polyclonal) | Millipore | (Millipore 12–371) RRID:AB_145840 | isotype control (1:100) |
| Antibody | H3K27me3 (rabbit monoclonal) | Abcam | ab6002, RRID:AB_305237 | ChIP (1:100) |
| Antibody | EZH2 (rabbit monoclonal) | Cell Signal Technology | # 5246 RRID:AB_10694683 | ChIP (1:100) |
| Antibody | DDK/FLAG (mouse monoclonal) | Merck | #F3165 RRID:AB_259529 | ChIP (1:100), WB (1:1000) |
| Antibody | Anti-FLAG M2 Magnetic Beads | Millipore | M8823 RRID:AB_2637089 | for protein purification |
| Recombinant DNA Reagent | pENTR11 | Invitrogen | K253520 | Plasmid (Shuttle vector for Gateway Cloning) |
| Recombinant DNA Reagent | pCW57.1 | Addgene | 41393 | Plasmid (Tet. Inducible lentiviral system - gateway cloning), used for generating TERT and TRF2 PTM induced expression system |
| Recombinant DNA Reagent | TRF2 shRNA | origene | TL308880 | Plasmid |
| Recombinant DNA Reagent | pSpCas9(BB)–2A-Puro (PX459) V2.0 | Addgene | #62988 RRID:Addgene_62988 | Plasmid |
| Recombinant DNA Reagent | Control shRNA Plasmid-A | Santa-cruz | sc-108060 | Plasmid |
| Recombinant DNA Reagent | TERC shRNA | Santa-Cruz | sc-106994-SH | Plasmid |
| Recombinant DNA Reagent | pCMV6-myc-DDK(FLAG)-TRF2 vector (TRF2 WT, R17H) | origene | RC223601 | Plasmid |
| Sequence-Based Reagent | GTR OLIGO | Human telomeric sequence | synthesised by Sigma | (TTAGGG)4 |
| Peptide, Recombinant Protein | histone H3 N-terminal peptide | Active Motif kit | Catalog No. 56100 | used as template for in vitro histone methyltransferase assay |
| Peptide, Recombinant Protein | EZH2 /EED/SUZ12/RbAp48/AEBP2 human | Sigma-Aldrich | cat no.SRP0381 | used for in vitro histone methyltransferase assay |
| Commercial Assay Or Kit | Histone H3 (tri-methyl K27) Quantification Kit (Colorimetric) | abcam | ab115072 | used for in vitro histone methyltransferase assay |
| Commercial Assay Or Kit | Histone H3 (K27) Methyltransferase Activity Quantification Assay Kit | abcam | ab113454 | histone assay buffer |
| Commercial Assay Or Kit | ROCHE TeloTAGGG Telomerase PCR ELISA | Roche | Cat. No. 11 854 666 910 | Telomeric Repeat Amplification Protocol Kit |
| Chemical Compound, Drug | Doxycycline | Sigma Aldrich | D9891 | 1 µg/mL |
| Chemical Compound, Drug | Puromycin | GIBCO | Catalog number A1113803 | 1 µg/mL |
| Software, Algorithm | UTRdb 2.0 | http://212.189.202.211/utrdb/index_107.html | | Used to design TERT 3' UTR specific primers |
| Other | Tetracycline-free Fetal Bovine Serum (FBS) | Tet System Approved FBS Clontech Laboratories, Inc. | #631106 | |
| Other | Stellaris FISH Probes, Human TERT with Quasar 670 Dye | LGC BiosearCh Technologies | VSMF-2411–5 | |

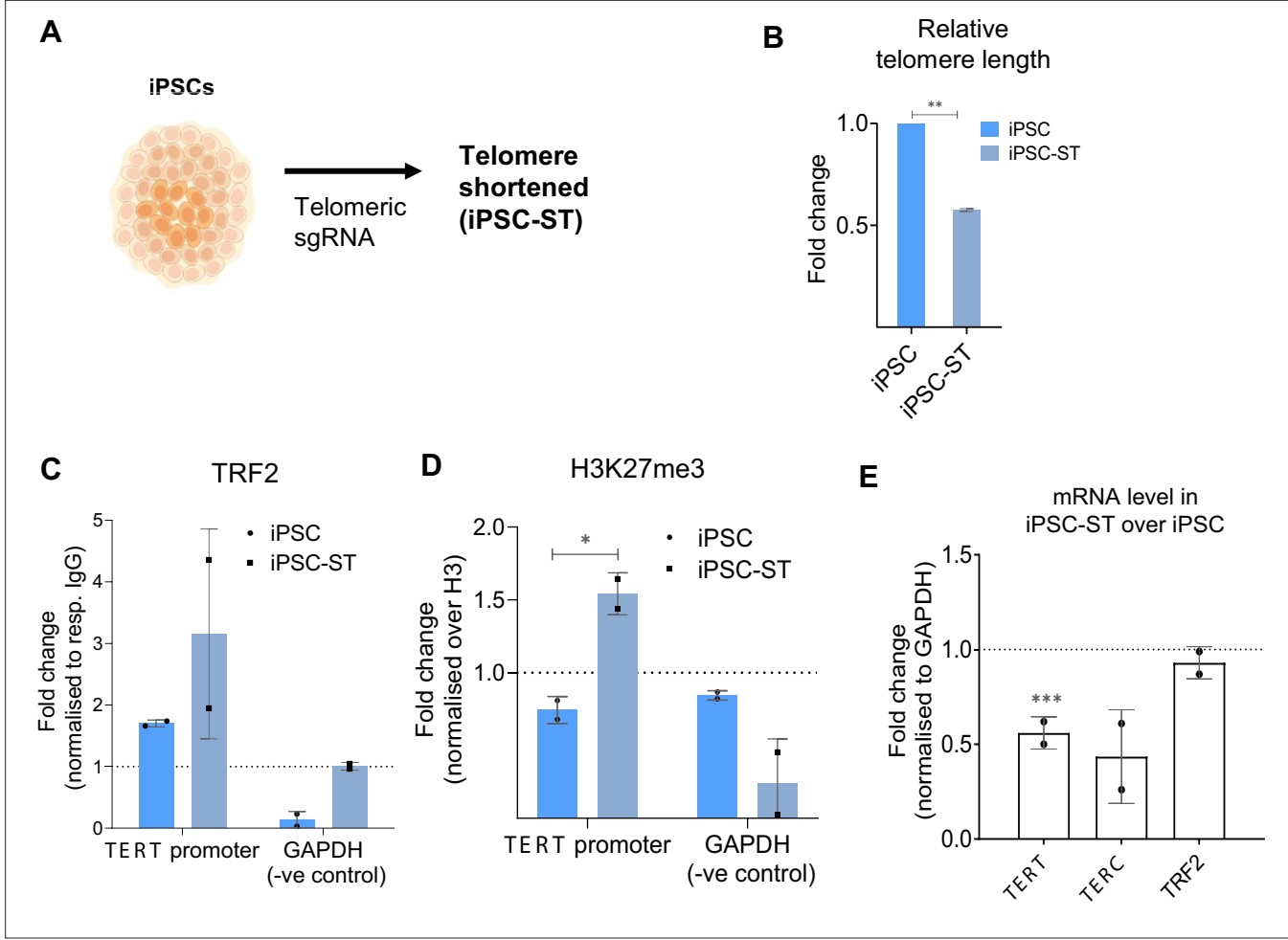

**Figure 11.** Telomere shortening in induced pluripotent stem cellss (iPSCs) increases TRF2 binding and represses *TERT*. (**A**) Scheme depicting generation of induced pluripotent stem cells (iPSCs) with shortened telomeres, (iPSC-ST) using telomere-specific sgRNA-guided CRISPR-Cas9 to trim telomeres. (**B**) Relative fold change in telomere length in iPSC-ST cells with respect to unaltered iPSC, determined by qPCR-based telomere length detection method. (**C–D**) ChIP followed by qPCR at the 0–300 bp *TERT* promoter (upstream of TSS) in iPSC-ST cells in comparison to unaltered iPSC, for TRF2 (**L**) and H3K27me3 (**M**); occupancy normalized to respective IgG or total Histone H3 (for H3K27me3). qPCR on the GAPDH promoter was used as a negative control in all cases. (**E**) mRNA levels for *TERT* (full-length exon 15/16 transcript), TERC (RNA component), and TRF2 in iPSC-ST over unaltered iPSC. All error bars represent ± SDs from the mean of 2 independent biological replicates of each experiment. p values are calculated by unpaired t-test (*p<0.05, **p<0.01, ***p<0.005, ****p<0.0001).

The online version of this article includes the following source data for figure 11:

**Source data 1.** Source data of all plots in *Figure 11*.

## Cancer cell lines

HT1080 (obtained from ATCC) and its derivative cell line HT1080-LT (received as a kind gift from Dr. J. Ligner's lab) were maintained in Minimum Eagle's Essential Medium (MEM) (Sigma-Aldrich) with 10% Fetal Bovine Serum (FBS) (Gibco). HCT116 *TP53* null (gift from Prof. Bert Vogelstein's lab), HCT116 WT, MDA-MB- 231, and HEK293T (purchased from NCCS, Pune), CRISPR CCR5 locus insert cell lines; along with their derivative cells (short or long telomeres) were grown in Dulbecco's Modified Essential Medium High Glucose (DMEM-high Glucose) (Sigma-Aldrich) with 10% FBS. Doxycycline (dox) Inducible TRF2 cells in HT1080 and Dox-HT1080 were cultured in MEM and dox Inducible TRF2 HCT116 and MDA-MB-231 and Dox-MDA-MB-231 cells were cultured in DMEM-High Glucose, all supplemented with 10% tetracycline-free Fetal Bovine Serum (FBS) (Clontech and DSS Takara). Cells were trypsinized and subcultured in desired culture vessels when at 90% confluency. All cells were grown in 5% $CO_2$, 95% relative humidity, and 37 °C culturing conditions. Authentication of cell lines was

carried out using STR profiling, and periodic PCR testing verified that no mycoplasma contamination was present.

## Generation of short or long-TL cellular model systems

1.  A longer telomere version of HT1080 fibrosarcoma cells (termed super-telomerase and referred to as HT1080-LT in the current paper) was generated by overexpression of telomerase (*TERT*) and telomerase RNA component (*TERC*) in the HT1080 unmodified cell line (termed as HT1080-ST in the current paper). It was received as a generous gift from Lingner Lab (*Cristofari and Lingner, 2006*). The two cell lines (HT1080 ST/LT) have been cultured and maintained independently of each other.

2.  HCT116 *TP53* null or knockout cells (termed LT here) were used as the parent cell line to generate another stable cell line with short telomere using commercial *TERC* knockdown plasmid (ST) and a scrambled control (LT) from Santa Cruz Biotechnology. Over multiple passaging under puromycin selection, telomere length reduction was achieved. This HCT116 ST/LT cell line pair was generated under *TP53* knockout (*TP53-/-*) background to prevent p53-regulated Siah1-mediated TRF2 degradation as reported earlier (*Sur et al., 2009*; *Fujita et al., 2010*).

3.  A long telomere length version of the MDA-MB-231 cell line was generated using the alternative lengthening of telomeres (ALT) mechanism (*Wright et al., 1996*). MDA-MB-231 cells were seeded and treated with GTR (guanine-rich terminal repeats) oligos at 3 uM concentration, in serum-free media for 24 hr. Post-treatment, the cells were kept in DMEM- High Glucose for 24 hr. Sequential treatment for six feedings (i.e. 12 days), increased telomere length by four-fold, as determined by Telomeric PNA-Flow cytometry (FACS), compared to its corresponding unmodified parental cell (MDA-MB-231 ST), being cultured simultaneously. These MDA-MB-231 ST/LT cells have been analysed as paired sets in all experiments.

4.  HCT116 cells were utilised to generate a short telomere length version utilizing CRISPR-Cas9-based telomere trimming. A telomeric sequence-specific sgRNA (tel-sgRNA) and *Sp*Cas9 expressing plasmid (PX459) were transfected into the cells to generate a telomere-shortened version (ST) of the cells. This mode of telomere shortening is quicker than that via *TERC* knockdown (as in point 2). Telomere length was quantified by FACS in comparison to the untransfected control HCT116 cells, after 3 days from transfection - post puromycin treatment (1 µg/mL) to select out transfected cells (with shorter telomeres). These HCT116 ST/LT cells (by telomere trimming) have been analysed as paired sets in all experiments.
    Telomeric sgRNA: GTTAGGGTTAGGGTTAGGGTTA(*Chen et al., 2013*)

5.  To generate the TL elongation followed by shortening model, the doxycycline-inducible *TERT* expression system, (stable cell generation discussed below), was used. The doxycycline-inducible system consists of cells with an integrated dox-*TERT* cassette, seeded in different culture flasks and maintained under three conditions: uninduced (no doxycycline), induced with doxycycline for 10 days, and post-induction withdrawal for defined time periods. Cells are harvested at multiple time points and analysed as paired samples. Briefly, the Dox-HT1080 cells were seeded in T25 flasks and treated with 1 µg/mL dox daily (one set being cultured without induction) and TL measured at time points Days 0/6/8/10/16 and 24. Beyond the first 10 days, dox treatment produced no further elongation (as seen on Day 16¯ in *Figure 5—figure supplement 1*). Cells were cultured till Day 24, after discontinuing dox at Day-10, for telomere shortening. Similar treatment was done for the Dox-MDA-MB-231 cellular modeland TL was measured on Days 0/4/8/10/12 and 14; cells were studied till Day 14 (after dox withdrawal at Day-10 and no further increase in TL with dox treatment, *Figure 6—figure supplement 1*) when substantial telomere shortening was noted.

6.  1300 bp region of *TERT* promoter starting from 48 bp downstream of TSS with *Gaussia* Luciferase reporter was procured from Genecopoeia-HPRM25711- PG04 (pEZX-PG04.1 vector) and inserted in HEK293T cells via CRISPR-Cas9 technology as discussed in *Sharma et al., 2021*. The telomeric sequence-specific sgRNA (tel-sgRNA) and *Sp*Cas9 expressing plasmid as mentioned for HCT116, these CRISPR-modified cells were transfected to generate a telomere-shortened version of the CRISPR cells for both unmutated *TERT* promoter and G- quadruplex (G4) disrupting mutant promoter forms, namely –124G>A and –146 G>A. Telomere length was quantified by FACS, after 3 days of transfection post-puromycin treatment to select out transfected cells (with shorter telomeres).

7.  Primary foreskin fibroblast cell line was obtained as a gift from Dr. Archana Singh's lab. They were grown in Dulbecco's Modified Essential Medium High Glucose (DMEM-high Glucose) (Sigma-Aldrich) with 10% FBS. Reprogramming was conducted using the CytoTune iPS 2.0 Sendai Reprogramming Kit (Thermo Scientific) according to the manufacturer's instructions. In

brief, fibroblast cells ($5\times10^5$) were transduced with reprogramming factor genes delivered by non-integrating Sendai viruses. The following day, cells were centrifuged at 200 g for 5 min at room temperature to remove the virus and then cultured for an additional two days. On the third day post-transduction, $0.5–2\times10^5$ cells were plated on Matrigel (Corning)-coated six-well plates and maintained in Essential 8 Medium (Gibco). The medium was changed daily, and cells were monitored for the emergence of colonies resembling embryonic stem (ES) cells. Between days 16 and 20 following transduction, colonies with flat, well-defined margins indicative of an ES-like phenotype were manually selected and expanded as iPSCs. These iPSC colonies were cultured on Matrigel-coated plates in Essential 8 Medium at 37 °C in a 5% CO2 environment, with daily media changes. During passaging, the typical split ratio was 1:5, and colonies were detached using ReLeSR (Stem Cell Technologies). To generate the telomere-shortened version of iPSCs, iPSCs were seeded in Matrigel-coated six-well plates and transfected with a telomeric sequence-specific sgRNA (tel-sgRNA) and *Sp*Cas9-expressing plasmid. 24 hr post-transfection, they were subjected to puromycin selection (0.4 µg/mL) for 5 days and monitored for any morphological or proliferation capacity changes.

Reagents catalog no:

| Reagent | Company | Catalog no |
| --- | --- | --- |
| CTS CytoTune-iPS 2.1 Sendai Reprogramming Kit | Thermo | A34546 |
| Essential 8 Medium | Thermo/Gibco | A1517001 |
| Matrigel | Corning | 354277 |
| ReLeSR | Stem Cell Technologies | 05872 |

Doxycycline (dox) inducible *TERT* and TRF2 post-translationally-modified (PTM) lentiviral stable cell generation

Doxycycline (dox) inducible *TERT* construct and TRF2 WT and R17H PTM variant construct- both in pCW57.1 backbone were individually transfected with 3rd generation lentiviral packaging plasmids (pRRE, pREV, and pMD2.G) in 4:3:3:1 ratio in HEK293 Lenti-X cells for viral particle generation. 12 hr post-transfection, media change was given and incubated in general culture conditions for 72 hr. Viral particles were then collected by filtering the media with a 0.45 µM syringe filter. This viral particle filtrate was then added along with fresh media and Polybrene (5 µg/mL) to HT1080 and MDA-MB-231 cells (for ind. *TERT* construct) and HT1080, HCT116 WT cells, and MDA-MB-231 cells (for ind. TRF2 variant construct); incubated for 24 hr. Post 24 hr, media containing viral particles was safely discarded and fresh media was added. Next, puromycin selection was given to the cells until the transduced population of cells could be stably maintained.

## Generation of TRF2 PTM variants using site-directed mutagenesis

The TRF2 post-translational modification variants were generated using site-directed mutagenesis by PCR method. The primers containing the required mutation (as documented in the table) were used to amplify the entire pCMV6 plasmid containing the TRF2-WT sequence. The PCR product was purified and transformed into DH5α *E. coli* cells. Then the mutated plasmid constructs were isolated and confirmed by Sanger sequencing.

| Mutation | Forward primer (FP) (5′–3′) | Reverse primer (RP) (5′–3′) |
| --- | --- | --- |
| K190R | CCAAGGACCCCAC AACTCAGAGGCTGA GAAATGA | TCATTTCTCAGCCTCT GAGTTGTGGGGTCCTTGG |
| T188N | TGTCCAAGGACCCCA CAAATCAGAAGCTGAGAAAT | ATTTCTCAGCTTCTGA TTTGTGGGGTCCTTGGACA |
| K176R | AAAAACAAAGAATTTG AAAAGGCTTCAAGAATTTTGA AAAAACATATGTCCAAGGAC | GTCCTTGGACAT ATGTTTTTTCAAAAT TCTTGAAGCCTTTTCAA ATTCTTTGTTTTT |
| R17H | CGGGCAGCTGGCC ACCGGGCGTCCCGC | GCGGGACGCCCG GTGGCCAGCTGCCCG |

## Generation of xenograft tumour in NOD-SCID mice

Tumour xenograft generation in NOD SCID mice was outsourced to Vivo Bio Tech. Ltd.

Telangana, India. 2.5 million cells of HT1080, and HT1080-LT each were injected subcutaneously in healthy male mice (4–6 weeks old) and monitored for tumour growth. Tumours were harvested by sacrificing mice when tumour growth reached an average volume of 600 mm³(± 100).

## Telomere length determination

Telomere length was determined using two methods - RT-PCR-based and Flow Cytometry-based. For flow cytometry-based evaluation using Agilent DAKO kit, $2 \times 10^6$ cells were divided into two fractions- for FITC-labelled Telomeric PNA probe (Agilent DAKO kit) and unlabelled control cells. The cell fractions were washed and further labelled with PI for whole DNA content as per the manufacturer's protocol. The fluorescence signal was acquired for at least 10,000 events on BD Accuri C6 flow cytometer. FCS files were then analyzed using FlowJo version 10.8.1 software. Fold change of telomere length relative to reference cell samples was calculated using Median Fluorescence signal intensity values of labelled and unlabelled sample preparations, according to the kit-provided formula.

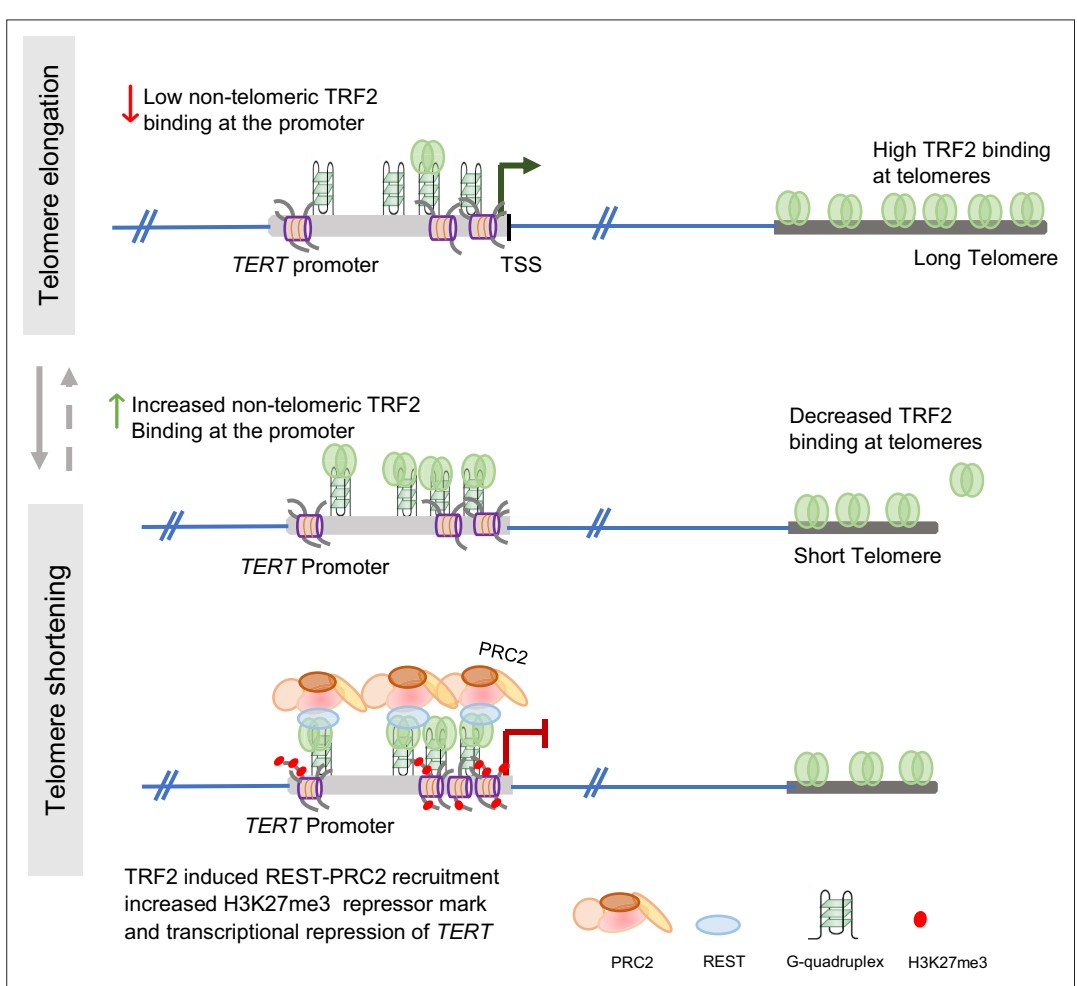

**Figure 12.** Telomere length controls human telomerase (*TERT*) expression through non-telomeric TRF2. Illustration of the telomere-dependent epigenetic modification of chromatin structure at the *TERT* promoter, resulting in upregulation or downregulation of *TERT* expression due to altered non-telomeric TRF2 binding at the promoter. Relatively long telomeres cells (top panel) have lower TRF2 binding at the *TERT* promoter, promoting permissive chromatin and upregulation of *TERT* transcription. Conversely, shorter telomere cells (bottom panel) with increased TRF2 binding at the *TERT* promoter recruit more REST-PRC2 epigenetic complex causing increased repressor histone H3K27 trimethylation deposition. This leads to a more closed chromatin state at the *TERT* promoter, suppressing its transcription.

For RT-PCR-based telomere length estimation, genomic DNA was isolated using the Wizard Genomic DNA Purification kit. At least two dilutions of genomic DNA in the range of 10 ng - 60 ng were prepared for each sample and qRT-PCR was set up for telomere-specific primers and single copy gene 36B4 specific primer from *O'Callaghan et al., 2008* and *Cawthon, 2002*, respectively (*Cawthon, 2002*; *O'Callaghan et al., 2008*). Ct value of telomeric primer was compared to respective sample 36B4 and then fold change over the control sample was calculated.

The primers used are (5' – 3'):

Tel F CGGTTTGTTTGGGTTTGGGTTTGGGTTTGGGTTTGGGTT
Tel R GGCTTGCCTTACCCTTACCCTTACCCTTACCCTTACCCT
36B4 F CAGCAAGTGGGAAGGTGTAATCC
36B4 R CCCATTCTATCATCAACGGGTACAA

## Telomerase activity using ELISA TRAP

ELISA TRAP was performed as described in *Sharma et al., 2021*. Briefly, $1 \times 10^6$ cells were lysed using mild lysis buffer CHAPS supplemented with RNase inhibitor to prevent loss of enzymatic activity (due to *TERC* degradation) and PIC to prevent protein degradation. Protein concentration in the lysate was estimated and diluted to 0.5 µg/µL concentration. 2 µg of protein was then used to set up telomerase repeat amplification protocol (TRAP) PCR (*Kim et al., 1994*). This was performed using a PCR master mix provided with ROCHE TeloTAGGG Telomerase PCR ELISA kit which uses 3' biotinylated telomeric sequence repeat primers, P1-TS primers. Telomerase from the lysate extends the P1-TS primer,, which in the next step acts as a template for amplicon generation using P1-TS primer and (TTAGGG)6 repetitive sequences- P2. After this PCR, dioxygenin (dig) tagged telomere-specific detection probes are hybridized to denatured PCR products. These tagged biotinylated PCR products are then bound to streptavidin-coated plates and quantified using anti-dig-Peroxidase conjugated antibody, which produces coloured product upon metabolizing substrate, TMB.

## ChIP (chromatin immunoprecipitation)

Relevant Primary antibodies (TRF2 Novus Biologicals #NB110-57130, REST Sigma Aldrich #17–641, EZH2 Cell Signal Technology # 5246, H3K27me3 Abcam #ab6002, H3 Abcam # ab1791, FLAG Merck #F3165) and suitable IgG (from Millipore, for Isotype control or Mock) were used to perform the ChIP assays as per protocol reported in *Mukherjee et al., 2018b*. Briefly, $3 \times 10^6$ cells were harvested and fixed with 1% formaldehyde for 10 min. Post washing with ice-cold PBS, the fixed cells were lysed in SDS-based lysis buffer with 2 X mammalian Protease Inhibitor Cocktail (mPIC) and subjected to chromatin fragmentation to an average size range of 200–300 bp using Bioruptor (Diagenode). 10% fragmented chromatin was processed as Input fraction via phenol-chloroform-isoamyl Alcohol (PCI) and Ethanol precipitation. Assay was performed with 3 µg of respective primary antibody incubated overnight on rotor at 4 °C followed by immune complex pulldown with salmon sperm DNA saturated Protein G Magnetic Dyna Beads. Subsequent washes with low salt, high salt and then LiCl buffers were performed and beads resuspended in TE (Tris-EDTA pH 8) for proteinase K treatment at 55 °C, 2 hr. Thereafter, PCI was used to phase-separate DNA in aqueous layer; separated fraction was incubated with equal volumes of isopropanol, 3 M sodium acetate and glycogen overnight at –20 °C. Pulldown DNA was then precipitated by centrifugation and pellet washed in 70% ethanol before air drying and resuspending in nuclease free water. Chromatin pulldown was validated by q-PCR analysis.

## Analysis of ChIP experiments

ChIP-q-PCR for TRF2, REST, and EZH2 chromatin pulldowns was performed with equal amount of DNA for each fraction, determined by Qubit HS DNA kit. Then, fold change was calculated over IgG (Mock) by using average Ct values. In case of histone ChIP assays too, equal amount of DNA from each histone ChIP and its respective total H3 were used in qPCR. Ct values for Histone mark and total H3 were normalised to 1% Input and thereafter fold change over total H3 was calculated.

## Transfections

Cells were seeded a day before transfection at 70% confluency in a six-well plate. 1.5 µg plasmid per well was transfected by complexing with FUGENE at 1:3 (DNA: Reagent) ratio following the

manufacturer's protocol. 12 hr post-transfection media change was given to the cells and later harvested for mRNA and protein analysis after 36 hr of media change.

## Real-time PCR for mRNA expression

Total RNA isolation was performed using TRIzol Reagent (Invitrogen, Life Technologies) as per the manufacturer's instructions. RNA quantification and cDNA preparation was done using the Applied Biosciences kit. Quantitative real-time PCR using SYBR Green-based method (DSS TAKARA) was employed to estimate relative transcript levels of mRNAs with GAPDH as housekeeping control gene for internal normalization. Average fold change was calculated by difference in internally normalised threshold cycles (Ct) between test and control samples. All experiments were performed in three independent biological replicates; with technical triplicates of each primer pair and sample combination for q-PCR analysis.

In HT1080 ST/LT, Dox-HT1080, and Dox-MDA-MB-231 cells with overexpression of *TERT* and *TERC* (for HT1080-LT), primer pair specific for *TERT* mRNA 3'UTR was implemented to capture the status of endogenous *TERT* expression in addition to standard primers spanning exon 15/16 (full length) and exon 7/8 (active reverse transcriptase) forms. 3'UTR *TERT* primers were designed with help from UTRdb 2.0.

mRNA primers (5' – 3'):

> *TERT* 3' UTR FP AATTTGGAGTGACCAAAGGTGT
> *TERT* 3' UTR RP TATTTTACTCCCACAGCACCTC
> *TERT* exon 15/16 FP GGGTCACTCAGGACAGCCCAG
> *TERT* exon 15/16 RP GGGCGGGTGGCCATCAGT
> *TERT* exon 7/8 FP GCGTAGGAAGACGTCGAAGA
> *TERT* exon 7/8 RP ACAGTTCGTGGCTCACCTG
> GAPDH FP    TGCACCACCAACTGCTTAGC
> GAPDH RP      GGCATGGACTGTGGTCATGAG
> TRF2 FP      CAGTGTCTGTCGCGGATTGAA
> TRF2 RP CATTGATAGCTGATTCCAGTGGT
> 18S FP    TTCGGAACTGAGGCCATGAT
> 18S RP    TTTCGCTCTGGTCCGTCTTG

## *TERT* mRNA FISH

Untreated or transfected cells were seeded on coverslip for this experiment and processed according to Stellaris RNA FISH protocol. Briefly, cells were fixed with fixation buffer for 10 min at RT. This was followed by PBS wash and permeabilization with 70% (v/v) ethanol. For hybridization, 125 nM of probe per 100 µl of Hybridization buffer was mixed and added onto parafilm surface in a humidified chamber. Coverslips were then placed on this drop with cells side down (post-wash in wash buffer A containing formamide), incubated in dark at 37 °C overnight. After incubation, the coverslips were transferred to fresh 1 ml Wash Buffer A for 30 min. This was followed by nuclear staining with DAPI in the same buffer, in the dark for 30 min maximum. Finally, the coverslip was washed in Wash Buffer B for 5 min and then mounted on the slide with anti-fade mounting reagent.

Post slide processing, fluorescence signal for Quasar 670 *TERT* RNA probe and DAPI were acquired on Leica SP8 confocal microscopy system. Implementing the default Rolling ball radius feature of open-source image analysis software Fiji ImageJ, background signal subtraction of each image frame was performed. Thereafter, signal from n=30 cell nucleus (manually marked) was quantified.

## Immunofluorescence microscopy

Adherent cells were seeded on coverslips at a confluency of ~70% before transfecting them with required TRF2 overexpression plasmids. At the harvesting time-point, cells were fixed using freshly prepared 4% Paraformaldehyde and incubated for 10 min at RT. Cells were permeabilised with 0.1% Triton X-100 (10 min at RT). The cells were treated with blocking solution (5% BSA in PBS+0.1% Tween 20 (PBST)) for 1 hr at RT. All the above steps were followed by three washes with ice-cold PBS for 5 min each. Post-blocking, cells were treated with anti-FLAG antibody (Merck #F3165, 1:1000 diluted in 5% PBST) and incubated overnight at 4 °C in a humid chamber. Post-incubation, cells

were washed alternately with PBS three times for 5 min and probed with secondary Ab mouse Alexa Fluor 488 (A11059, 1:3000) for 2 hr at RT, in the dark. Cells were washed again with PBS three times and mounted with Prolong Gold anti-fade reagent with DAPI. Images were taken on Leica TCS-SP8 confocal microscope.

## TRF2 protein purification

$1.5 \times 10^6$ HEK293T cells (~60% confluency) were transfected with 3 µg of TRF2- WT or TRF2-R17H overexpression pCMV6 plasmid. 24 hr post-transfection, cells were treated with G418 selection for 36–48 hr. Post this period of overexpression under selection pressure, the cells were harvested and resuspended in around 3–4 mL ice-cold TBS (150 mM NaCl, 50 mM Tris-Cl, pH7.4) with 2 X mammalian Protease Inhibitor Cocktail (mPIC). The cells were then sonicated to lyse open at 15 s ON/ 30 s OFF cycles for 5 min (until turbidity was reduced), at 4 °C. Cell debris was then separated by centrifuging at 13,000 rpm, 5 min, at 4 °C. M8823 M2 FLAG Dynabeads were prepared for incubation with the cell lysate as follows: 40 µL of the bead slurry was washed/ equilibrated twice in TBS by loading on 1.5 mL tube magnetic stand, kept on ice. This equilibrated bead solution is then added to the cell lysate and incubated overnight at 4 °C.

Post incubation of lysate with M2 FLAG bead for binding, the supernatant solution was removed and beads (bound to tube on magnetic stand) were washed based on step gradient purification. Two washes with 1 mL each of 150 mM, 300 mM, 500 mM, and 1 M NaCl containing TBS were given to remove non-specific proteins bound to beads. All the wash fractions along with the supernatant and purified bead-bound TRF2 fractions were then analysed by running a 10% SDS-PAGE and staining with Coomassie Brilliant blue.

## Histone H3 methyltransferase activity quantification assay

1 µg of purified histone H3 N-terminal peptide (procured as a component of Active Motif kit Catalog No. 56100) was incubated overnight (12–14 hr), at 37 °C in the presence of 8 µM (working concentration) S-adenosyl-L-methionine, 0.6 µg purified PRC2 complex of proteins (cat no.SRP0381) and in the presence/absence of M2 FLAG bead-bound TRF2 protein (WT or R17H variants) in 25 µL of a histone assay buffer (component of ab113454 kit). After that, the protocol of ab115072 –Histone H3 (tri-methyl K27) quantification kit was followed. Briefly, 25 µL of supernatant of the reaction mix was pipetted, added to the anti-H3K27me3 antibody-coated strip wells with 25 µL of antibody buffer (provided with kit ab115072) and incubated at room temperature for 2 hr; the wells were covered properly by Parafilm M to prevent evaporation. Post incubation, the wells were washed thrice with kit-provided wash buffer. Diluted detection antibody (as directed by kit) was then added to the wells and incubated for 1 hr at room temperature on an orbital shaker at 100 rpm. Next, the wells were washed thrice after aspirating the antibody solution. The suggested amount of Colour Developer solution was added to the wells and incubated in the dark at room temperature for 2–10 min. The colour development (blue) was monitored across the wells of sample and standard solution. Once the colour in the standard well containing a higher concentration developed a medium blue colour, 50 µL of Stop Solution was added to all wells to stop the enzymatic reaction. Upon stopping the reaction, a yellow colour develops, and the absorbance reading at 450 nm was noted within 15 min on a microplate reader. Fold change in trimethylation over reference condition is determined by the following formula:

$$\text{Fold change of Tri} - \text{methylation} = (\text{Treated (tested) sample OD} - \text{blank OD})/(\text{Untreated (control) sample OD} - \text{blank OD})$$

Blank in case of experiment was the reaction mix without PRC2 complex and TRF2 protein forms.

## Western blot analysis

Western blot analysis for checking the integrity of the purified bead-bound protein was also carried out. An equal volume of bead (5 µL) was loaded for both TRF2-WT and TRF2-R17H on a 10% SDS-PAGE and transferred on polyvinylidene difluoride (PVDF) membranes (Biorad). Membranes were blocked with 5% skimmed milk and then probed suitably with primary antibodies- anti-TRF2 antibody (Novus Biological), and anti-FLAG antibody (Merck). Anti-rabbit and anti-mouse HRP-conjugated secondary antibodies from CST, respectively were used. Blots were developed using BioRad ClarityMax HRP chemiluminescence detection kit and images acquired on the LAS500 GE ChemiDoc system.

### *Gaussia*-luciferase assay

Secreted *Gaussia* luciferase signal in the CRISPR-modified short telomere and control cells (WT, –124G>A and –146 G>A -ST/LT) was quantified using Promega *Gaussia* luciferase kit according to the manufacturer's protocol. The luminescence signal (arbitrary unit, a.u.) was normalised to total protein for each condition.

## Quantification and statistical analysis

All experiments were performed with biological replicates, and data are presented as the mean ± standard deviation (SD) unless otherwise stated. Statistical significance was determined using appropriate tests based on experimental design.

For comparisons between groups where samples were derived independently (e.g. comparisons between each parental cell line and its corresponding short or long telomere-altered counterpart, or between control and TRF2-overexpressing conditions), an unpaired two-tailed Student's t-test was used. For experiments in which the same set of biological replicates was subjected to different treatments or time points (e.g. doxycycline induction and post-withdrawal), a paired two-tailed Student's t-test was applied.

For comparisons across more than two time points or conditions (e.g. telomere length analysis or expression dynamics during doxycycline treatment), one-way ANOVA followed by Tukey's Honestly Significant Difference (HSD) post-hoc test was used to identify statistically significant differences between groups.

For experiments involving two independent variables (e.g. *Guassia* luciferase assay across multiple conditions and time points, or in vitro histone methyltransferase activity under different treatments), two-way ANOVA was performed to assess the interaction effects. Where applicable, post-hoc comparisons were conducted to identify significant pairwise differences.

Statistical significance is indicated as follows: $*p<0.05$; $**p<0.01$; $***p<0.005$; $****p<0.0001$. All statistical analyses and data visualization were performed using GraphPad Prism version 8.0.2.

## Acknowledgements

The authors thank Balam Singh for the management of the laboratory and resources. We are thankful to all members of the S.C. laboratory for their suggestions/inputs and the IGIB Core Imaging Facility for its service facility. We acknowledge assistance from Dr. Arnab Mukhopadhyay and his lab members (National Institute of Immunology) for their cooperation. Research fellowships to ASG, SVM, DS, RD, JJ, SD, AKM, SSR, PP, SS, AKB, DK, AS, SS, SB (CSIR), and M.Y.(UGC) are acknowledged. This work was supported by Wellcome Trust/DBT India Alliance Fellowship (IA/S/18/2/504021). Support from the Council of Scientific and Industrial Research (CSIR) and the Department of Biotechnology (DBT) to S.C. are also acknowledged. The funders had no role in study design, data collection and analysis, decision to publish, or preparation of the manuscript.

## Additional information

### Funding

| Funder | Grant reference number | Author |
| --- | --- | --- |
| Wellcome Trust/DBT India Alliance | IA/S/18/2/504021 | Shantanu Chowdhury |
| Council of Scientific and Industrial Research, India | | Shantanu Chowdhury |
| Department of Biotechnology, Ministry of Science and Technology, India | | Shantanu Chowdhury |

| Funder | Grant reference number | Author |
|--------|------------------------|--------|

The funders had no role in study design, data collection and interpretation, or the decision to submit the work for publication. For the purpose of Open Access, the authors have applied a CC BY public copyright license to any Author Accepted Manuscript version arising from this submission.

## Author contributions

Antara Sengupta, Resources, Data curation, Formal analysis, Validation, Investigation, Visualization, Methodology, Writing – original draft, Project administration; Soujanya Vinayagamurthy, Subhajit Dutta, Validation, Investigation, Methodology; Dristhi Soni, Rajlekha Deb, Ananda Kishore Mukherjee, Jushta Jaiswal, Mukta Yadav, Investigation, Methodology; Shalu Sharma, Sulochana Bagri, Shuvra Shekhar Roy, Akshay Sharma, Suman Saurav, Rajender K Motiani, Resources, Methodology; Priya Poonia, Amit Kumar Kumar Bhatt, Investigation; Ankita Singh, Resources, Validation; Divya Khanna, Resources; Shantanu Chowdhury, Conceptualization, Supervision, Funding acquisition, Project administration, Writing – review and editing

## Author ORCIDs

Antara Sengupta https://orcid.org/0000-0002-0974-9961
Soujanya Vinayagamurthy https://orcid.org/0000-0003-1465-9925
Subhajit Dutta https://orcid.org/0000-0003-2390-7492
Shuvra Shekhar Roy https://orcid.org/0000-0001-6005-2767
Amit Kumar Kumar Bhatt https://orcid.org/0009-0002-8455-685X
Rajender K Motiani https://orcid.org/0000-0002-8971-9008
Shantanu Chowdhury https://orcid.org/0000-0001-7185-8408

## Ethics

The animal experiments were performed in accordance with guidelines and recommendations laid out by the institutional animal ethics committee (IAEC) at CSIR-Institute of Genomics and Integrative Biology-Delhi, India after due clearance was awarded to Dr. Shantanu Chowdhury (IGIB/IAEC/11/28/2020).

Reviewer #1 (Public review): https://doi.org/10.7554/eLife.104045.3.sa1
Author response https://doi.org/10.7554/eLife.104045.3.sa2

---

# Additional files

## Supplementary files
MDAR checklist

## Data availability
All data generated during this study are included in the article and figure supplements.

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
