## [Editor Report · eLife Assessment]

The authors of this **important** study investigate how telomere length regulates hTERT expression via non-telomeric binding of the telomere-associated protein TRF2. They conclusively show that TRF2 binding to long telomeres results in a reduction in its binding to the hTERT promoter, while short telomeres restore TRF2 binding in the hTERT promoter, recruiting repressor complexes like PRC2, and suppressing hTERT expression. There is **convincing** support for the claims and the findings should be of broad interest for cell biologists and those working in fields where telomeres alter function, such as cancer and aging.

---

## [Referee Report · Reviewer #1 (Public review)]

Summary:

The authors in this study extensively investigate how telomere length (TL) regulates hTERT expression via non-telomeric binding of the telomere-associated protein TRF2. They conclusively show that TRF2 binding to long telomeres results in a reduction in its binding to the hTERT promoter. In contrast, short telomeres restore TRF2 binding in the hTERT promoter, recruiting repressor complexes like PRC2, and suppressing hTERT expression. The study presents several significant findings revealing a previously unknown mechanism of hTERT regulation by TRF2 in a TL-dependent manner

Strengths:

(1) A previously unknown mechanism linking telomere length and hTERT regulation through the non-telomeric TRF2 protein has been established, strengthening our understanding of telomere biology.

(2) The authors used both cancer cell lines and iPSCs to showcase their hypothesis and multiple parameters to validate the role of TRF2 in hTERT regulation.

(3) Comprehensive integration of the recent literature findings and implementation in the current study.

(4) In vivo validation of the findings.

(5) Rigorous controls and well-designed assays have been used.

Comments on current version:

The current version of the manuscript has addressed all the reviewers' concerns to the best of its ability. However, understanding the limitations of the authors, exploring ALT cell lines for the current mechanism would be desirable in the future.

---

## [Author Response]

The following is the authors’ response to the original reviews.

**Reviewer #1 (Public review):**
Summary:The authors in this study extensively investigate how telomere length (TL) regulates hTERT expression via non-telomeric binding of the telomere-associated protein TRF2. They conclusively show that TRF2 binding to long telomeres results in a reduction in its binding to the hTERT promoter. In contrast, short telomeres restore TRF2 binding in the hTERT promoter, recruiting repressor complexes like PRC2, and suppressing hTERT expression. The study presents several significant findings revealing a previously unknown mechanism of hTERT regulation by TRF2 in a TL-dependent mannerStrengths:(1) A previously unknown mechanism linking telomere length and hTERT regulation through the non-telomeric TRF2 protein has been established strengthening the telomere biology understanding.(2) The authors used both cancer cell lines and iPSCs to showcase their hypothesis and multiple parameters to validate the role of TRF2 in hTERT regulation.(3) Comprehensive integration of the recent literature findings and implementation in the current study.(4) In vivo validation of the findings.(5) Rigorous controls and well-designed assays have been use.Weaknesses:(1) The authors should comment on the cell proliferation and morphology of the engineered cell lines with ST or LT.

The cell proliferation and morphology of the engineered cells were monitored during experiments. With a doubling time within 16-18 hours, all the cancer cell line pairs used in the study were counted and seeded equally before experiments.

No significant difference in morphology or cell count (before harvesting for experiments) was noted for the stable cell lines, namely, HT1080 ST-HT1080 LT, HCT116 p53 null scrambled control-HCT116 p53 null hTERC knockdown.

MDAMB 231 cells which were treated with guanine-rich telomere repeats (GTR) over a period of 12 days, as per the protocol mentioned in Methods. Due to the alternate day of GTR treatment in serum-free media followed by replenishment with serum-supplemented media, we noted that cells would undergo periodic delay in their proliferation (or transient arrest) aligning with the GTR oligo-feeding cycles and appeared somewhat larger in comparison to their parental untreated cells.

Next, the cells with Cas9-telomeric sgRNA mediated telomere trimming were maintained transiently (till 3 days after transfection). During this time, no significant change in morphology or cell proliferation was observed in any of the cell lines, namely HCT116 or HEK293T Gaussia Luciferase reporter cells. iPSCs were also monitored. However, no change in morphology or cellular proliferation was observed during the 5 days post-transfection and antibiotic selection.

(2) Also, the entire study uses engineered cell lines, with artificially elongated or shortened telomeres that conclusively demonstrate the role of hTERT regulation by TRF2 in telomere-length dependent manner, but using ALT negative cell lines with naturally short telomere length vs those with long telomeres will give better perspective. Primary cells can also be used in this context.

The reviewer correctly highlights (as we also acknowledge in the Discussion) that our study primarily utilizes engineered cell lines with artificially elongated or shortened telomeres. We agree that using ALT-negative cells with naturally short versus long telomeres would provide additional perspective. However, a key challenge in this experimental setup is the inherent variation in TRF2 protein levels among these cell types—a parameter central to our hypothesis. Comparing observations across such non-isogenic cell line pairs presents experimental limitations as these would require extensive normalization for multiple factors and introduce additional complexities, which would be difficult to interpret with clarity.

We had also explored primary cells, specifically foreskin fibroblasts and MRC5 lung fibroblasts, as suggested by the reviewer. However, we encountered two significant challenges. To achieve a notable telomere length difference of at least 20%, these primary cells had to undergo a minimum of 25 passages. During this period, we observed a substantial decline in their proliferation capacity and an increased tendency toward replicative senescence. Additionally, we noted a significant reduction in TRF2 protein levels as the primary cells aged, consistent with findings from Fujita K et al., 2010 (Nat Cell Biol.), which reported p53-induced, Siah-1-mediated proteasomal degradation of TRF2. Due to these practical limitations, we focused on cancer cell lines with respective isogenic backgrounds, ensuring a controlled experimental framework. On the other hand, this opens new avenues for future research to explore broader implications. Investigating other primary cell types that may not present these challenges could be a valuable direction for future studies.

(3) The authors set up time-dependent telomere length changes by dox induction, which may differ from the gradual telomere attrition or elongation that occurs naturally during aging, disease progression, or therapy. This aspect should be explored.

In this study, we utilized a Doxycycline-inducible hTERT expression system to modulate telomere length in cancer cells, aiming to capture any gradual changes that might occur upon steady telomerase induction or overexpression—an event frequently observed in cancer progression. We monitored telomere length and telomerase activity at regular intervals (Supplementary Figure 2), noting a gradual increase until a characteristic threshold was reached, followed by a reversal to the initial telomere length.

While this model provides interesting insights in context of cancer cells, it does not replicate the conditions of aging or therapeutic intervention. We agree that exploring telomere length-dependent regulation of hTERT in normal aging cells is an important avenue for future research. Investigating TRF2 occupancy on the hTERT promoter in response to telomere length alterations through therapeutic interventions—such as telomestatin or imetelstat (telomerase inhibitors) and 6-thio-2’-deoxyguanosine (telomere damage inducer)—would provide valuable insights and warrants further exploration.

(4) How does the hTERT regulation by TRF2 in a TL-dependent manner affect the ETS binding on hTERT mutant promoter sites?

In our previous study (Sharma et al., 2021, Cell Reports), we have experimentally demonstrated that GABPA and TRF2 do not compete for binding at the mutant hTERT promoter (Figure 4M-R). Silencing GABPA in various mutant hTERT promoter cells did not increase TRF2 binding. While GABPA has been reported to show increased binding at the mutant promoter compared to the wild-type (Bell et al., 2015, Science), no telomere length (TL) sensitivity has been noted yet. In the current manuscript we show that telomere alterations in hTERT mutant cells (that do not form promoter G-quadruplex) does not significantly affect TRF2 occupancy at the promoter, reinforcing our earlier findings that G-quadruplex formation is crucial for TRF2 recruitment. Since TRF2 binding is not affected this would not impact GABPA binding. Therefore change in TL is unlikely to influence ETS binding by GABPA.

(5) Stabilization of the G-quadruplex structures in ST and LT conditions along with the G4 disruption experimentation (demonstrated by the authors) will strengthen the hypothesis.

We agree with the reviewer’s suggestion that stabilizing G-quadruplex (G4) structures in mutant promoter cells under ST and LT conditions would further strengthen our hypothesis. From our ChIP experiments on hTERT promoter mutant cells following G4 stabilization with ligands, as reported in Sharma et al. 2021 (Figure 5G), we observed that TRF2 occupancy was regained in the telomere-length unaltered versions of -124G>A and -146G>A HEK293T Gaussia luciferase cells (referred to as LT cells in the current manuscript).

(6) The telomere length and the telomerase activity are not very consistent (Figure 2A, and S1A, Figure 4B and S3). Please comment.

In this study, we employed both telomerase-dependent and independent methods for telomere elongation.

HT1080 model: Telomere elongation resulted from constitutive overexpression of hTERC and hTERT, leading to a direct correlation with telomerase activity.

HCT116 (p53-null) model: hTERC silencing in ST cells, a known limiting factor for telomerase activity, resulted in significantly lower telomerase activity and a 1.5-fold telomere length difference.

MDAMB231 model: Guanine-rich telomeric repeat (GTR) feeding induced telomere elongation through recombinatorial mechanisms (Wright et al., 1996), leading to significant telomere length gain but no notable change in telomerase activity.

HCT116 Cas9-telomeric sgRNA model: Telomere shortening occurred without modifying telomerase components, resulting in a minor, insignificant increase in telomerase activity (Figure 2A, S1).

Regarding xenograft-derived HT1080 ST and LT cells (Figure 4B, S3), the observed variability in telomere length and telomerase activity may stem from infiltrating mouse cells, which naturally have longer telomeres and higher telomerase activity than human cells. Since in the reported assay tumour masses were not sorted to exclude mouse cells, using species-specific markers or fluorescently labelled HT1080 cells in future experiments would minimize bias. However, even though telomere length and telomerase activity assays cannot differentiate for cross-species differences, mRNA analysis and ChIP experiments performed specifically for hTERT and hTERC mRNA levels, TRF2 occupancy, and H3K27me3 enrichment on hTERT promoter (Figure 4B–E) strongly support our conclusions.

(7) Please comment on the other telomere-associated proteins or regulatory pathways that might contribute to hTERT expression based on telomere length.

The current study provides experimental evidence that TRF2, a well-characterized telomere-binding protein, mediates crosstalk between telomeres and the regulatory region of the hTERT gene in a telomere length-dependent manner. Given the observed link between hTERT expression and telomere length, it is likely that additional telomere-associated proteins and regulatory pathways contribute to this regulation.

The remaining shelterin complex components—POT1, hRap1, TRF1, TIN2, and TPP1—may play crucial roles in this context, as they are integral to telomere maintenance and protection (Stewart J et al., 2012 Mutat Res.). Additionally, several DNA damage response (DDR) proteins, which interact with telomere-binding factors and help preserve telomere integrity, could potentially influence hTERT regulation in a telomere length-dependent manner (Longhese M, 2008 Genes & Development). However, direct interactions or regulatory roles would require further experimental validation. Another group of proteins with potential relevance in this mechanism are the sirtuins, which directly associate with telomeres and are known to positively regulate telomere length, undergoing repression upon telomere shortening (Amano H et al., 2019 Cell Metabolism, Amano H, Sahin E 2019 Molecular & Cellular Oncology). Notably, SIRT1 has been reported to interact with telomerase (Lee SE et al., 2024, Biochem Biophys Res Commun.), while SIRT6 has been implicated in TRF2 degradation (Rizzo et al. 2017) and telomerase activation (Chen J et al. 2021, Aging) . Given their roles in telomere homeostasis, sirtuins may serve as key mediators of telomere length-dependent hTERT regulation.

Based on this suggestion, we have included the above in Discussion.

**Reviewer #2 (Public review):**
Summary:Telomeres are key genomic structures linked to everything from aging to cancer. These key structures at the end of chromosomes protect them from degradation during replication and rely on a complex made up of human telomerase RNA gene (hTERC) and human telomerase reverse transcriptase (hTERT). While hTERC is expressed in all cells, the amount of hTERT is tightly controlled. The main hypothesis being tested is whether telomere length itself could regulate the hTERT enzyme. The authors conducted several experiments with different methods to alter telomere length and measured the binding of key regulatory proteins to this gene. It was generally observed that the shortening of telomere length leads to the recruitment of factors that reduce hTERT expression and lengthening of telomeres has the opposite effect. To rule out direct chromatin looping between telomeres and hTERT as driving this effect artificial constructs were designed and inserted a significant distance away and similar results were obtained.Overall, the claims of telomere length-dependent regulation of hTERT are supported throughout the manuscript.Strengths:The paper has several important strengths. Firstly, it uses several methods and cell lines that consistently demonstrate the same directionality of the findings. Secondly, it builds on established findings in the field but still demonstrates how this mechanism is separate from that which has been observed. Specifically, designing and implementing luciferase assays in the CCR5 locus supports that direct chromatin looping isn't necessary to drive this effect with TRF2 binding. Another strength of this paper is that it has been built on a variety of other studies that have established principles such as G4-DNA in the hTERT locus and TRF2 binding to these G4 sites.Weaknesses:The largest technical weakness of the paper is that minimal replicates are used for each experiment. I understand that these kinds of experiments are quite costly, and many of the effects are quite large, however, experiments such as the flow cytometry or the IPSC telomere length and activity assays appear to be based on a single sample, and several are based upon two maximum three biological replicates. If samples were added the main effects would likely hold, and many of the assays using GAPDH as a control would result in significant differences between the groups. This unnecessarily weakens the strength of the claims.

We appreciate the reviewer’s recognition of the resource-intensive nature of our experiments, and we are confident in the robustness of the observed results. Due to the project’s timeline constraints and the need for consistency across experiments, we have reported findings based on 3 biological replicates with appropriate statistical analysis.

Regarding the fibroblast-iPSC model, we would like to clarify that we have presented data from two independent biological replicates, each consisting of a fibroblast and its derived iPS cell pair, rather than a single sample. Additionally, the Tel-FACS assays involved analysing at least 10,000 events, ensuring statistical significance in all cases.

Another detail that weakens the confidence in the claims is that throughout the manuscript there are several examples of the control group with zero variance between any of the samples: e.g. Figure 2K, Figure 3N, and Figure 6G. It is my understanding that a delta delta method has been used for calculation (though no exact formula is reported and would assist in understanding). If this is the case, then an average of the control group would be used to calculate that fold change and variance would exist in the group. The only way I could understand those control group samples always set to 1 is if a tube of cells was divided into conditions and therefore normalized to the control group in each case. A clearer description in the figure legend and methods would be required if this is what was done and repeated measures ANOVA and other statistics should accompany this.

The above point has been raised by the reviewer in the 'Recommendations for Authors' section as well. We have addressed it in detail in that section, citing each figure where the reviewer noted a concern regarding the lack of variance. Changes made in the manuscript have also been highlighted there.

We would like to clarify that, throughout the manuscript, fold changes were previously calculated independently for each biological replicate by normalizing treated conditions to their corresponding control (untreated or Day 0) sample within the same replicate. This means that the control group is normalized to 1 individually in each replicate, resulting in an apparent lack of variance in the control when plotted. The normalization was not performed using an averaged control value across replicates. As such, the absence of visible variance in the control group reflects the normalization method rather than a true lack of variability in the underlying data.

In the revised version of the manuscript, we have carefully considered the reviewer’s comments and applied changes wherever appropriate. For example (detailed response in the ‘Recommendations for Authors’ section), in datasets where two distinct stable cell lines are compared (e.g., HT1080 ST/LT and HCT p53-null ST/LT), unpaired statistical analysis is more appropriate. Hence, we have updated these panels accordingly and indicated the statistical methods used in the figure legends and Methods section. However, in experiments where cells were indeed seeded separately and subsequently subjected to experimental conditions—representing paired samples—we have chosen not to make any changes. A clearer description of this procedure has, however, been added to the Methods and figure legends to ensure full transparency.

We believe this approach accurately reflects the experimental design, appropriately addresses the reviewer’s concerns regarding variance and statistical analysis, and ensures clarity and rigor in data reporting.

A final technical weakness of the paper is the data in Figure 5 where the modified hTERT promoter was inserted upstream of the luciferase gene. Specifically, it is unclear why data was not directly compared between the constructs that could and could not form G4s to make this point. For this reason, the large variance in several samples, and minimal biological replicates, this data was the least convincing in the manuscript (though other papers from this laboratory and others support the claim, it is not convincing standalone data).

We appreciate the reviewer's thoughtful feedback on the presentation of the luciferase assay data in Figure 5. The data for the wild-type hTERT promoter (capable of forming G4 structures) was previously reported in Figure 2G-K. To avoid redundancy in data presentation, we initially chose to report the results of the mutated promoter separately. However, we recognize that directly comparing the wild-type and mutated promoter constructs within the same figure would provide clearer context and strengthen the interpretation of the results. In light of this, we have updated Figure 5 in the revised manuscript to include the data for both constructs, ensuring a more comprehensive and informative comparison.

The second largest weakness of the paper is formatting.When I initially read the paper without a careful reading of the methods, I thought that the authors did not have appropriate controls meaning that if a method is applied to lengthen, there should be one that is not lengthened, and when a method is applied to shorten, one which is not shortened should be analysed as well. In fact, this is what the authors have done with isogenic controls. However, by describing all samples as either telomere short or telomere long, while this simplifies the writing and the colour scheme, it makes it less clear that each experiment is performed relative to an unmodified. I would suggest putting the isogenic control in one colour, the artificially shortened in another, and the artificially lengthened in another.Similarly, the graphs, in general, should be consistent with labelling. Figure 2 was the most confusing. I would suggest one dotted line with cell lines above it, and then the method of either elongation or shortening below it. I.e. HT1080 above, hTERC overexpression below, MDAMB-231 above guanine terminal repeats below, like was done on the right. Figure 2 readability would also be improved by putting hTERT promoter GAPDH (-ve control) under each graph that uses this (Panel B and Panel C not just Panel C). All information is contained in the manuscript but one must currently flip between figure legends, methods, and figures to understand what was done and this reduces clarity for the reader.

We thank the reviewer again for their thoughtful suggestions regarding figure formatting and colour coding to improve clarity. We fully understand the rationale for proposing separate colours for unmodified, telomere-shortened, and telomere-lengthened groups, as this could make the experimental design more immediately apparent. However, after careful consideration, we believe that implementing this change across all figures may unintentionally reduce clarity in other aspects (presented in other figures) of the data presentation. This is further explained below.

Specifically, applying three distinct colours throughout would make it harder to visually track key biological trends—such as changes in chromatin occupancy—across different models. For instance, the same colour could represent opposing regulatory patterns in distinct contexts (e.g., upregulation in one model and downregulation in another), which will make these figures difficult to understand. We feel that maintaining a consistent colour scheme based on telomere status—i.e., long telomeres (LT) vs short telomeres (ST)—across figures facilitates better comparison of biological outcomes across different experimental systems.

Nevertheless, to address the reviewer’s concern about clarity in experimental design, we have added more detailed descriptions of the methodology and model systems used, in both the Methods and figure legend sections. These updates aim to make it easier for the reader to follow which groups serve as isogenic controls versus modified samples, without disrupting the consistency of data visualization.

We hope this strikes a balance between improving clarity and preserving the interpretability of the broader biological trends presented in our manuscript.

Please note, we have incorporated the reviewer’s suggestion to indicate details of model generation for HT1080 and MDAMB 231 cell lines in Figure 2. To quote the reviewer,

“I would suggest one dotted line with cell lines above it, and then the method of either elongation or shortening below it. I.e. HT1080 above, hTERC overexpression below, MDAMB-231 above guanine terminal repeats below, like was done on the right.”

We have also put hTERT promoter GAPDH (-ve control) under each graph and not at the end of Panel C in Figure 2, as suggested by reviewer.

**Reviewer #1 (Recommendations for the authors):**
(1) Please check for grammatical errors throughout the manuscript.

We have gone through the manuscript thoroughly, checked and corrected it for grammatical errors if and where detected.

(2) Please use both the FACS and qPCR-based assays to check telomere length in all the experiments to strengthen the observations.

We would like to thank the reviewer for this valuable suggestion. We confirm that both FACS- and qPCR-based assays were performed to assess telomere length in our experiments. In the original submission, we chose to present primarily the FACS-based data in the main figures. This decision was based on the inherent differences in the measurement principles of the two methods, which can lead to discrepancies in the reported fold changes. We were concerned that presenting both datasets side by side in the main figures might lead to confusion for readers who are not directly familiar with the nuances of telomere length assays.

However, in light of the reviewer’s suggestion, we have now included the qPCR-based data as Supplementary Figure 1A, and updated the manuscript text and figure legends accordingly to reflect this addition.

(3) Correct the labeling in the legend (Figure 2).

We have corrected legend of Figure 2. Thanks to the reviewer for pointing it out.

(4) In Figure 6B, why TRF WT condition have higher hTERT expression than the UT condition?

We thank the reviewer for noting that the hTERT mRNA levels, as estimated by FISH in Figure 6B, appear slightly higher in TRF2 WT overexpressing HT1080 cells compared to the untransfected (UT) condition. Specifically, the average mean intensity values (a.u.) were 53 for UT and 57 for WT. Although this difference was not statistically significant, we acknowledge the reviewer's observation. Currently, we do not have a clear explanation for this small, non-significant variation.

Importantly, using the same FISH-based method, we observed a significant upregulation of hTERT mRNA levels upon TRF2 R17H overexpression compared to both UT and TRF2 WT conditions, supporting our key conclusions.

Additionally, qRT-PCR analysis of hTERT mRNA levels in cells stably expressing TRF2 WT (induced by doxycycline) consistently showed a significant downregulation compared to the uninduced (equivalent to UT in the microscopy experiments) state. These results were robust and reproducible across three different cell lines, including HT1080. Consistently, TRF2 R17H expression led to significant upregulation of hTERT mRNA levels upon induction.

Together, these complementary findings strengthen the validity of our observations.

(5) In telomere length between ST and LT in Fig. 5B significant? (especially the right panel -146G>A).

We consistently worked with approximately 20–30% telomere shortening in HEK293 cells across all three cell types (WT promoter, -124G>A, and -146G>A), as this range was reproducibly achieved within the experimental timeframe without risking excessive telomere trimming. The reported telomere length differences are based on FACS analysis of more than 10,000 events per condition, providing strong statistical significance. Importantly, while the absolute differences in telomere length may appear modest, their biological impact is evident in the distinct cellular characteristics observed between ST and LT cell pairs.

**Reviewer #2 (Recommendations for the authors):**
As mentioned above it was somewhat unclear why so many instances of control groups had no variance between them. A more complete reporting of the formulas used to calculate the results, and methods (if samples were divided from a single source into different conditions) would be appreciated.

We thank the reviewer for their valuable and detailed feedback. The instances where the control groups appeared to lack variance were mainly mRNA data (Figure 2D, 3G,3N), luciferase activity (Figure 2K), and in vitro methyltransferase activity (Figure 6G). We shall try to categorically address them all.

In Figure 2D, for the MDA-MB-231 GTR oligo and HCT116 telomere trimming datasets, the untreated cells were seeded separately and subsequently used to generate the treated conditions within the same experiment. Thus, these two datasets represent paired experimental conditions. Fold changes were calculated independently for each replicate (paired samples), and the fold changes across replicates were plotted. Because the control group serves as a common baseline within each pair and fold changes are normalized individually, minimal variance appears across controls. Given the experimental design, we believe no change is necessary for these panels. However, we have provided additional clarification regarding the calculation formulas and sample handling in the Methods section to avoid any ambiguity.

For the ST/LT versions in HT1080 and HCT p53-null background cells, while each replicate could technically be treated as paired, these could be treated as four distinct stable cell lines. Hence, we agree it would be appropriate to apply unpaired statistical analysis for these datasets. We have updated the plots accordingly and described the statistical methods in detail in the figure legends and Methods section.

Figure 3G and 3N depict the doxycycline-induced cells which follow the design: untreated and dox-treated conditions were seeded from the same batch of cells into separate flasks and treated differently. Hence, these are also paired cases, and fold changes were calculated per replicate before plotting. Therefore, we believe no changes are necessary for these panels. However, we have provided more details regarding sample handling in the Methods section to avoid any ambiguity.

In Figure 2K, previously we had plotted fold change in luciferase activity over short telomere (ST) cells, for each independent biological replicates. However, to address the reviewer’s concern of not showing variance in control group, we have now plotted the luminescence signal (normalised over total protein). We have also updated Figure 5E accordingly, and also included WT promoter data along with the mutant cell line data- as was suggested in public reviewer’s comment.

In Figure 6G, as each replicate of the in vitro methyltransferase activity used different batches of purified protein, there are inherent batch differences that were accounted for by normalizing each replicate internally. Fold changes were then determined for each replicate separately, as previously described. The fold changes across replicates were plotted, and significance between different conditions was tested using two-way ANOVA. To address the reviewer’s comment to show variance in the control, we have now plotted individual replicates.

We believe these revisions, along with the expanded methods clarification, will fully address the reviewer's concerns and accurately reflect the experimental design and statistical analysis applied.

Many times, in the manuscript a / is used to indicate both directions. For example: "Genes distal from telomeres (for instance 60 Mb from the nearest telomere) were activated/repressed in a TL-dependent way"... "Resulting increase/decrease in non-telomeric promoter-bound TRF2 affected gene expression". For readability, either this can be replaced with a directionless word like altered, changed, etc, or the writer can list both directions.

We thank the reviewer for the careful reading and thoughtful suggestions. In the manuscript, we have used the ‘/’ symbol to indicate opposing directions, followed by the word ‘respectively’ to relate these directions to their corresponding outcomes, wherever appropriate. However, as rightly pointed out, certain sentences would benefit from alternative constructions for improved clarity and readability. We have therefore reviewed the manuscript and revised such sentences, making minor modifications wherever necessary, as outlined below.

We found hTERT was transcriptionally altered depending on telomere length (TL).

Notably, another conceptually distinct mechanism of TL-dependent gene regulation was reported which influenced genes spread throughout the genome: expression of genes distal from telomeres (for instance 60 Mb from the nearest telomere) was altered in a TL-dependent way, but without physical telomere looping interactions.

Second, the shortening or elongation of telomeres led to the release or sequestration of telomeric TRF2, respectively, thereby increasing or decreasing the availability of TRF2 at non-telomeric promoters and affecting gene expression.

A non-necessary, but potentially extra convincing experiment to perform would be to use a combination of light-activated, or ligand-activated cas9 telomere trimming and guanine terminal repeat additions in the same cell line. Like the dox experiments, this would show over time how altering telomere length alters the recruitment of heterochromatin factors and hTERT levels. Executing the experiment this way would be more definitive as it does not rely on changing hTERT itself. Authors do already have examples that support their claims.

We thank the reviewer for suggesting this additional experiment (reviewer mentions as non-necessary), which would indeed provide valuable insights into the relationship between telomere length, heterochromatin factor recruitment, and hTERT levels. While we recognize the potential of this approach, due to constraints on resources, we are currently unable to execute this experiment. However, we believe that the existing data presented in the manuscript already supports our conclusions effectively.